# Engineering mammalian cells for detection and treatment of cardiac injury

Yaqing Si [1,2,3,13], Yuxuan Fan [1,2,4,13], Leo Scheller [5,11], Bozhidar-Adrian Stefanov [5,12], Jian Lv[6,7], Zhihua Wang [6,7], Mingqi Xie [1,2,8,9] & Martin Fussenegger [5,10]

## Abstract

Early detection of myocardial abnormalities or other ischemic heart diseases is critical for effective treatment. Here, we aimed to engineer a cell-based system to sense cardiac troponin I (cTnI), an early marker of acute myocardial infarction (AMI), and respond by releasing a thrombolytic agent. To detect cTnI, we engineered a chimeric troponin receptor (TropR) that contains extracellular single-chain variable fragments (scFvs) and signals via intracellular domains of interleukin 6 receptor subunit beta (IL6RB), epidermal growth factor receptor (EGFR), fibroblast growth factor receptor 1 (FGFR1), fibroblast growth factor receptor 2b (FGFR2b) or vascular endothelial growth factor receptor 2 (VEGFR2) that are associated with cardioprotective signaling. cTnI-dependent TropR functionality was confirmed in human embryonic kidney (HEK)-derived cell lines as well as iPSC-derived cardiomyocytes, and enabled rapid, reversible, tunable control of gene expression via synthetic-signaling-specific promoters. We then constructed monoclonal cell lines for cTnI-induced secretion of the thrombolytic protein tenecteplase (TNK), together with an off-switch triggered by FDA-approved doxycycline. We selected a clone, designated Cardio-Protect, whose sensitivity was optimized to detect human AMI-relevant cTnI levels. To validate thrombolytic efficacy, we established an ex vivo blood culture system and show that alginate-microencapsulated CardioProtect cells triggered complete lysis of fibrin clots in a strict cTnI-inducible, doxycycline-repressible manner. This closed-loop strategy serves as a proof-of-concept for using cell therapy in the early detection and treatment of AMI.

**Keywords** Cardiac Troponin; Acute Myocardial Infarction; Synthetic Receptors; Designer Cells; Thrombolysis
**Subject Categories** Biotechnology & Synthetic Biology; Cardiovascular System

## Introduction

Ischemic heart disease is a leading cause of global death, accounting for 9.14 million deaths in 2019 (Vos et al, 2020). In the United States, the acute myocardial infarction (AMI) subtype occurs in over 1 million patients per year, among which almost 50% of cases are fatal (Anderson and Morrow, 2017; Kunadian and Gibson, 2012). AMI is typically triggered by rupture or erosion of an atherosclerotic coronary plaque, resulting in the exposure of circulating blood to the highly thrombogenic core and matrix materials. This may lead to a totally occluding thrombus, resulting in the fatal stage of ST-elevation myocardial infarction (STEMI) (Anderson and Morrow, 2017). Even from the point of onset, fibrin clots can cause cardiac ischemic injury and concomitant cTnI release into the bloodstream. To reduce the likelihood of cardiovascular death, the recommended treatment option is rapid initiation of thrombolytic therapy (such as injection of recombinant tenecteplase (TNK)—a 527-amino-acid recombinant tissue-type plasminogen activator that selectively converts thrombus-bound plasminogen to the serine protease plasmin (Kunadian and Gibson, 2012; Logallo et al, 2015)), followed by immediate referral for coronary angiography at a facility where percutaneous coronary intervention (PCI) is available (Anderson and Morrow, 2017; Kunadian & Gibson, 2012; Logallo et al, 2015). However, the timely treatment of AMI and other cardiac dysfunctions is severely hampered by the absence of specific signs of discomfort at the early stages of heart injury, even though many cardiac biomarkers begin to accumulate in the circulation immediately (Valensi et al, 2011; McDonnell et al, 2009). In fact, half of all AMI deaths occur within the first hour after the appearance of symptoms, before the patient is able to reach the hospital (Kunadian and Gibson, 2012; Benedict et al, 1995). Thus, strategies capable of enabling both early detection of cardiac injury and rapid administration of thrombolytic agents are urgently required to reduce mortality rates among AMI patients (McDonnell et al, 2009).

To address this issue, we aimed to apply synthetic biology-inspired cell engineering principles to construct "therapeutic" designer cells as a proof of concept to investigate the feasibility of

[1]Westlake Laboratory of Life Sciences and Biomedicine, Hangzhou, Zhejiang, China. [2]Key Laboratory of Growth Regulation and Translational Research of Zhejiang Province, School of Life Sciences, Westlake University, Hangzhou, Zhejiang, China. [3]School of Basic Medical Sciences, Fudan University, Shanghai, China. [4]College of Life Sciences, Zhejiang University, Hangzhou, Zhejiang, China. [5]Department of Biosystems Science and Engineering, ETH Zurich, Klingelbergstrasse 48, CH-4056 Basel, Switzerland. [6]Shenzhen Key Laboratory of Cardiovascular Disease, Fuwai Hospital Chinese Academy of Medical Sciences, Shenzhen, China. [7]State Key Laboratory of Cardiovascular Disease, Fuwai Hospital, National Center for Cardiovascular Diseases, Chinese Academy of Medical Sciences and Peking Union Medical College, Beijing, China. [8]Institute of Basic Medical Sciences, Westlake Institute for Advanced Study, Hangzhou, Zhejiang, China. [9]School of Engineering, Westlake University, Hangzhou, Zhejiang, China. [10]Faculty of Science, University of Basel, Klingelbergstrasse 48, CH-4056 Basel, Switzerland. [11]Present address: Leibniz Institute for Immunotherapy, Regensburg, Germany. [12]Present address: Institute of Cell Biology, University of Berne, Berne, Switzerland. [13]These authors contributed equally: Yaqing Si, Yuxuan Fan. ✉E-mail: xiemingqi@westlake.edu.cn; fussenegger@bsse.ethz.ch

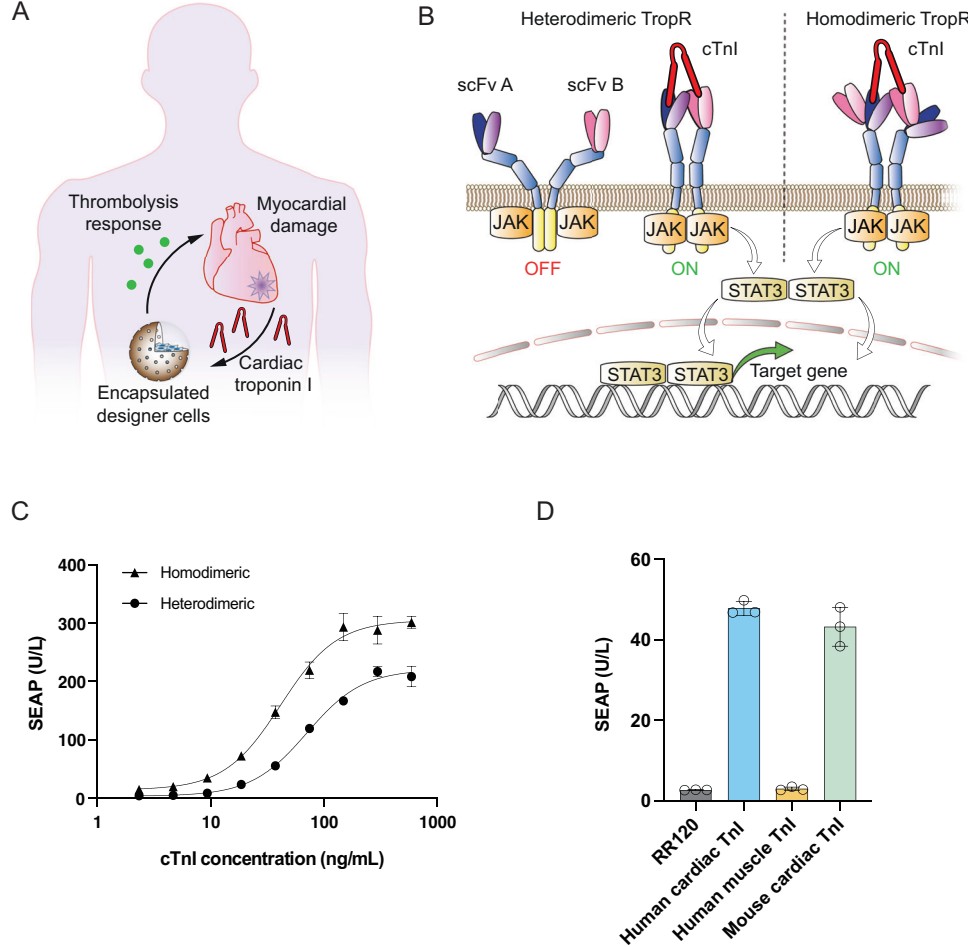

**Figure 1. Engineering of a chimeric TropR receptor for cell-based treatment of acute myocardial infarction.**

(A) Designer cells for closed-loop control of cTnI detection and in situ initiation of a therapeutic response. Upon cardiac injury, cardiac troponin I (cTnI) is released into the bloodstream as a specific biomarker of acute myocardial infarction (AMI). Implanted therapeutic cells designed to respond to systemic cTnI surges by secreting thrombolytic drugs could substantially improve the survival rate of AMI patients. One option for such implantation therapy would be encapsulation into a semipermeable membrane such as alginate-poly-L-lysine-alginate to avoid physical contact between the engineered cells and the patient's own tissues, thereby preventing undesired immune responses and other safety issues while allowing flexible exchange of nutrients and metabolites between the encapsulated cells and the host's bloodstream to enable rapid detection of disease markers (e.g., cTnI) and release of therapeutic agents (e.g., thrombolytics). (B) Engineering of different TropR receptor architectures for cTnI-dependent target gene expression. The chimeric TropR receptor comprises an extracellular scFv domain for cTnI binding, a transmembrane scaffolding domain derived from the erythropoietin receptor and an intracellular signal transduction domain of the interleukin-6 receptor (IL-6RB). Upon cTnI binding, receptor dimerization triggers intracellular JAK/STAT3 (Janus kinase/signal transducer and activator of transcription) signaling and activates gene expression from promoters containing STAT3-specific response elements. In heterodimeric TropR designs, two different scFvs binding different cTnI epitopes are placed on different TropR polypeptide subunits, whereas in homodimeric TropR, these scFvs are daisy-chained within the same molecule. (C) TropR-mediated cTnI-dependent target gene expression. HEK-293 cells were co-transfected with vectors encoding heterodimeric cTnI receptors (pLeo1058 and pLeo1059) or homodimeric receptor variant (pLeo1060), a constitutive STAT3-expression vector (pLS15) and a synthetic STAT3-specific SEAP expression vector (pTS566). Different concentrations of human cTnI were added to the cultivation medium, and SEAP levels in the culture supernatants were quantified at 24 h after cTnI stimulation. (D) TropR specificity for different troponin isoforms. HEK-293 cells were co-transfected with vectors encoding homodimeric cTnI receptors (pLeo1060), a constitutive STAT3-expression vector (pLS15) and a synthetic STAT3-specific SEAP expression vector (pTS566) before cultivation in culture medium supplemented with different troponin isoforms (human cardiac cTnI, 50 ng/mL; human slow-twitch muscle TnI, 200 ng/mL; murine cardiac cTnI, 50 ng/mL). Administration of reactive red 120 (RR120, 1 μg/mL) instead of troponin to transfected cells was used as a negative control. SEAP levels in culture supernatants were quantified at 24 h after cTnI stimulation. Data in (C, D) are presented as mean ± SD, n = 3 individual experiments. Source data are available online for this figure.

simultaneously treating and preventing AMI in the future (Fig. 1A). Cardiac troponin I (cTnI), a cardiomyocyte-specific intracellular protein that is only released into the circulation following myocardial necrosis or ischemia (Westermann et al, 2017; Reichlin et al, 2009), is the gold standard clinical marker for AMI (Anderson and Morrow, 2017; McDonnell et al, 2009). However, no natural receptor for cTnI exists. Thus, to allow human cells to respond to

cTnI, we first engineered a chimeric troponin receptor (TropR) capable of binding extracellular cTnI to trigger activation of intracellular signaling and initiation of target gene expression from synthetic promoters containing response elements for endogenous signaling-specific transcription factors. We then confirmed TropR functionality in various human cell types, constructed a human cell line capable of cTnI-dependent TNK secretion, and selected a clone

(named CardioProtect) having optimal sensitivity to detect human AMI-relevant cTnI levels. As a further step towards a potential treatment scenario in humans, we designed a thrombolysis model using human blood, and demonstrated that cTnI-stimulated alginate-encapsulated CardioProtect cells could cause complete dissolution of CaCl₂-induced AMI-like clots in human blood. Our findings suggest that synthetic biology-inspired designer cell therapies based on the implantation of encapsulated CardioProtect into human AMI patients could have translational potential to substantially advance cardiovascular healthcare.

# Results

## Engineering of a chimeric troponin receptor TropR

To engineer TropR, we capitalized on two single-chain variable fragment (scFv) antibodies selected to bind two different epitopes of cTnI (Conroy et al, 2012; Fukunaga and Tsumoto, 2013). Thus, following a chimeric design blueprint for synthetic receptors that allow mammalian cells to initiate customized genetic responses to various extracellular signals of interest (Scheller et al, 2018), we fused cTnI-specific scFvs (denoted 2B12 and A2, respectively) to a transmembrane scaffolding domain derived from the erythropoietin receptor (EpoR) and an intracellular signal transduction domain of the interleukin-6 receptor (Silver and Hunter, 2010). Thus, binding of cTnI by 2B12 and A2 is expected to trigger receptor dimerization, activation of JAK/STAT3-signaling and initiation of target gene expression from STAT3-specific promoters (Fig. 1B). In this canonical receptor architecture (Scheller et al, 2018), each scFv domain would be placed on a separate polypeptide subunit, thus forming a heterodimeric receptor architecture that requires two separate transcription units for expression in mammalian cells (Fig. 1B, left). When developing systems for cell-based therapies, however, compactness is a key factor to achieve increased treatment safety and persistence in vivo (Monteys et al, 2021) and to comply with size limitations of various gene delivery vectors. In fact, expression of heterodimeric TropR would require two episomal vectors (pLeo1058, pLeo1059; Table S1), each containing a separate transcription unit of at least 3 kb in size, whereas homodimeric TropR (pLeo1060; Table S1) can be designed to constitute transgene cargos of less than 3.8 kb. Therefore, we created a one-component expression system by designing a homodimeric receptor with the 2B12 and A2 scFvs daisy-chained at the N-terminus facing the extracellular space (Fig. 1B, right). Importantly, the linker between both scFv domains was carefully chosen to be short, so that the distance between the cTnI epitopes remains smaller than the distance between complementarity-determining regions (CDRs). In addition, these homodimeric receptors may show increased sensitivity by potentially binding two cTnI proteins simultaneously, thereby gaining in avidity. When tested in human embryonic kidney cells for proof of concept, both homodimeric and heterodimeric TropR architectures showed dose-dependent activation of STAT3-specific gene expression after exposure of transfected cells to different external cTnI doses. After quantification of human placental secreted alkaline phosphatase (SEAP) as the reporter in cell culture, the homodimeric variant demonstrated a higher dynamic range of fold-induction at the same cTnI concentration (Fig. 1C). Using flow-cytometric analysis, we observed similar expression levels of heterodimeric and homodimeric TropR (Appendix Fig. S1). This rules out major differences in expression efficiency of these different receptor constructs upon transient transfection in HEK-293 cells. These results show that functional receptors could be created despite exchanging the extracellular domains, thus highlighting the modular nature of this approach. Importantly, this TropR receptor was specific for cardiac troponin and remained insensitive to the skeletal muscle isoform (Fig. 1D), which could be an essential parameter for AMI-related treatments.

## Validation of TropR functionality in different human cell types

Next, we tested TropR functionality in mesenchymal stem cells (MSCs)—a clinically relevant cell type for myocardial infarction therapy (Golpanian et al, 2016; Karpov et al, 2017). cTnI-dependent activation of STAT3-regulated gene expression was achieved upon transient TropR transfection (Fig. 2A), showing comparable troponin sensitivity and fold-change to those previously characterized in HEK-293 cells (Fig. 1C). cTnI sensitivity was also conferred to human iPSC-derived cardiomyocytes (Fig. 2B). Since the TropR receptor was designed to also trigger endogenous signaling dynamics of its host cell in parallel with ectopic transgene expression upon stimulation, it is important that receptor activation can be rapidly halted as soon as the cTnI trigger is withdrawn. For example, it is known that transient activation of JAK/STAT3-signaling is required to trigger cardioprotective effects of MSC-based therapies (Shabbir et al, 2010; Niyaz et al, 2015), but permanent elevation of STAT3 activity is associated with the formation of cancer (Gu et al, 2020). Here, we observed a significant increase in reporter protein levels at 24 h after cTnI induction, but when the culture supernatant was changed to cTnI-free medium, no further increase in accumulated reporter protein levels was detected over the next 48 h (Fig. 2C). Gene expression was resumed upon reintroducing cTnI-containing medium. Thus, TropR-dependent gene expression indeed appeared to be cTnI-specific (Fig. 2C).

## Optimization of TropR to sense human AMI-relevant cTnI concentrations

Though TropR showed functionality in various human cell types, it is important that its troponin sensitivity matches the clinically relevant systemic range reported for AMI patients in the clinic. The current TropR variant responded well to peak cTnI levels of 100 ng/mL, which have been reported in human patients at 24 h post AMI (Mahajan & Jarolim, 2011) (Figs. 1C and 2A), but it would be desirable for a cell-based biosensor to show sensitivity to lower troponin levels in order to detect this molecular marker of AMI incidence at the earliest possible time point (i.e., ideally not later than within the first hour after a patient start to feel first signs of discomfort) (McDonnell et al, 2009). To improve cTnI sensitivity, we replaced the interleukin-6 receptor B (IL-6RB)-derived intracellular signal transduction domain of TropR by FGFR1-derived intracellular signal transduction domain (FGFR1$_{int}$) (Reichhart et al, 2016), FGFR2b-derived intracellular signal transduction domain (FGFR2b$_{int}$) (Liao et al, 2013) or EGFR-derived intracellular signal transduction domain (EGFR$_{int}$)

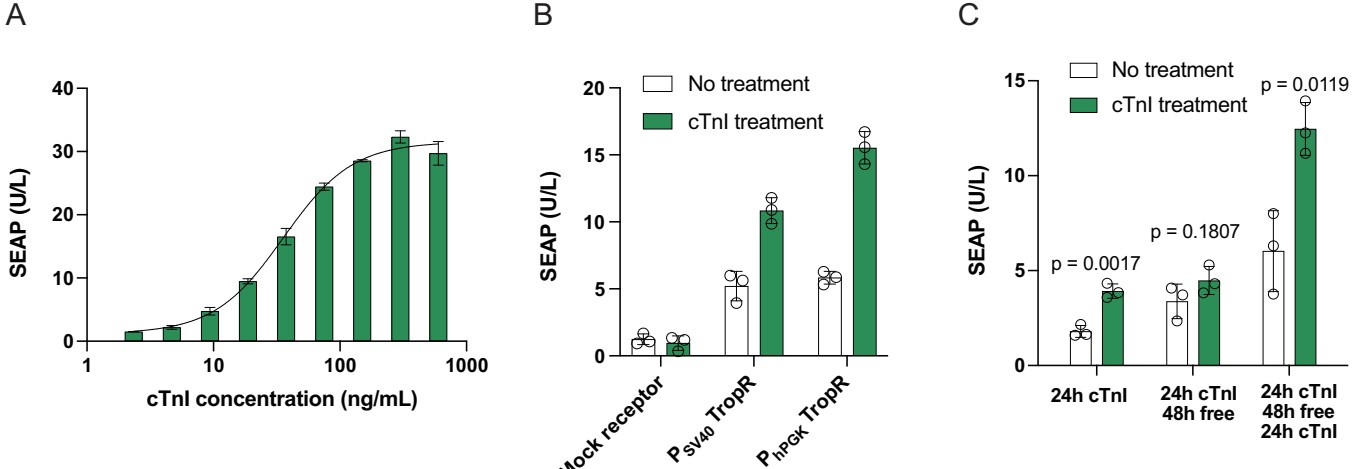

**Figure 2. TropR functionality in clinically relevant human cells.**

(A) TropR-mediated cTnI sensitivity in human mesenchymal stromal cells. hMSC-hTERT cells were co-transfected with vectors encoding homodimeric cTnI receptors (pLeo1060), a constitutive STAT3-expression vector (pLS15) and a synthetic STAT3-specific SEAP expression vector (pTS566) before different concentrations of human cTnI were added to the cultivation medium. SEAP levels in culture supernatants were quantified at 24 h after cTnI stimulation. (B) cTnI-dependent target gene expression in human iPSC-derived cardiomyocytes. iCell Cardiomyocytes were transfected with TropR expression vectors driven by different promoters ($P_{SV40}$, pLeo1060; $P_{hPGK}$, pLeo1120), a constitutive STAT3-expression vector (pLS15) and a synthetic STAT3-specific SEAP expression vector (pTS566) before human cTnI (100 ng/mL) was added to the cultivation medium. Transfection of an RR120-responsive GEMS receptor encoding vector (pLeo619) instead of TropR was used as a mock control. SEAP levels in culture supernatants were quantified at 24 h after cTnI stimulation. (C) Reversibility of cTnI-dependent target gene expression in human iPSC-derived cardiomyocytes. pLeo1060/pLS15/pTS566-transfected iCell Cardiomyocytes were cultivated for 4 days while human cTnI levels in the culture medium were successively switched between 0 and 100 ng/mL. SEAP levels were measured at different time points. Data in (A–C) are presented as mean ± SD, $n = 3$ individual experiments. Statistical significance was determined with Student's two-tailed $t$-test. Source data are available online for this figure.

(Greulich et al, 2005), three different signal transduction domains capable of activating MAPK signaling, as well as a VEGFR-derived intracellular signal transduction domain (VEGFR2$_{int}$) (Abhinand et al, 2016) for activation of endogenous calcium signaling (Fig. 3A). We also chose a secreted *Oplophorus gracilirostris* luciferase variant containing a IgK-derived signal peptide (NanoLuc) as a highly sensitive reporter protein whose luciferase signal is linearly proportional to the effective protein amount produced in mammalian cells (England et al, 2016). When tested in cells co-transfected with a synthetic MAPK-regulated transcription factor TetR-Elk1, cardiac troponin could regulate dose-dependent target gene expression from a synthetic TetR-specific promoter (Fig. 3B). Among the three MAPK-dependent variants tested, TropR containing FGFR2b$_{int}$ (designated TropR$_{FGFR2b}$) showed the highest cTnI sensitivity with an almost linear correlation to reporter gene expression in the range of 1–10 ng/mL (Fig. 3B). The VEGFR2$_{int}$-containing TropR variant successfully activated NFAT-specific promoters (Appendix Fig. S2A), but did not substantially differ from STAT3-based variants in terms of overall cTnI sensitivity, as shown in Figs. 1C and 2A. The same type of dose-response relationship was obtained irrespective of whether secreted NanoLuc (Fig. 3B; Appendix Fig. S2A) or SEAP was used as the target gene (Appendix Fig. S2B), suggesting that optimization principles may consistently apply across different isogenic experimental setups. Furthermore, we show that ectopic overexpression of various TropR variants had no significant impact on the activity of promoters that are known to respond to the cells' native JAK/STAT3 (Appendix Fig. S3A) or MAPK-signaling

pathways (Appendix Fig. S3B). Thus, we used the FGFR2b$_{int}$ domain for all further studies (pLeo1164, $P_{SV40}$-TropR$_{FGFR2b}$-pA).

Next, we created a monoclonal CardioReport cell line by stable integration of TropR$_{FGFR2b}$, TetR-Elk1 and a TetR-driven NanoLuc expression vector into human embryonic kidney cells. CardioReport cells retained the linear correlation between extracellular cTnI concentration and reporter gene expression within the critical range of 5–50 ng/mL (Fig. 3C), with an exposure time of 6 min being sufficient to activate the system (Fig. 3D). Furthermore, the system showed a reversible cTnI-dependent activation profile, where target gene expression was rapidly attenuated as soon as cTnI was removed from the medium in vitro (Fig. 3E). Notably, the synthetic transcription factor TetR-Elk1 allows for doxycycline-repressible target gene expression, a feature that could be exploited as a safety switch to enable external intervention during emergency situations in therapeutic settings or to "reboot" the therapeutic system for repetitive usage. In fact, we showed that administration of doxycycline could abrogate transcriptional activity triggered by a high cTnI concentration of 100 ng/mL (Fig. 3F).

## Closed-loop control of troponin-triggered thrombolysis

Thrombolytic therapy has revolutionized treatment options and survival rates of AMI patients (Kunadian and Gibson, 2012). At present, TNK is the first-line thrombolytic drug for emergency treatment of myocardial infarction if immediate percutaneous coronary intervention (PCI) cannot be performed and should be administered at the earliest time point following acute symptom

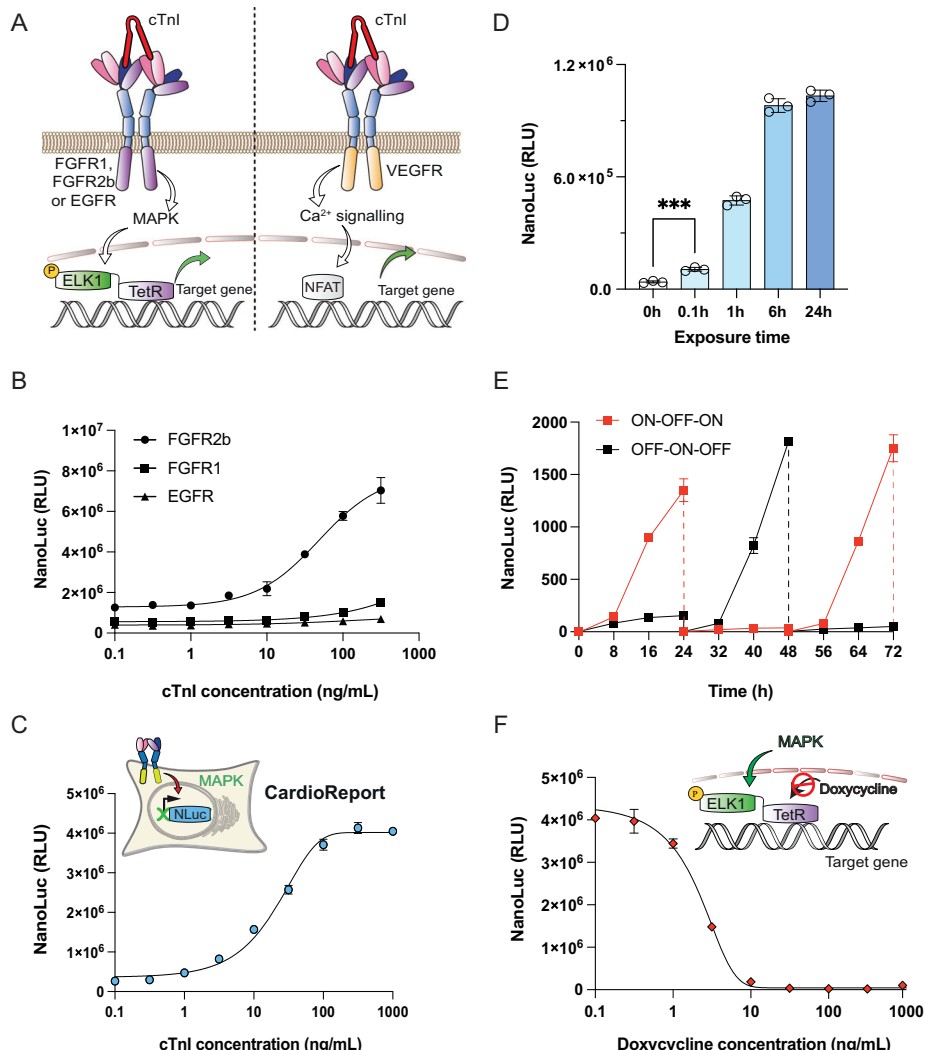

**Figure 3.  Optimization of TropR sensitivity by swapping intracellular signaling domains.**

(A) Rerouting TropR to trigger other intracellular signaling pathways. To achieve cTnI-triggered MAPK signaling, the intracellular interleukin-6 receptor (IL-6RB) domain of TropR was exchanged to FGFR1, FGFR2b or EGFR for activation of target gene expression from a synthetic TetR-ELK1-specific promoter. To achieve cTnI-triggered Ca$^{2+}$ signaling, a VEGFR intracellular signaling domain was used to activate target gene expression from synthetic NFAT-specific promoters. (B) Gene expression control by cTnI-triggered MAPK signaling. HEK-293 cells were co-transfected with a constitutive TetR-Elk1-expression vector (Mkp37), a synthetic TetR-specific NanoLuc expression vector (pSYQ377) and a TropR expression vector containing FGFR2b- (pLeo1164), FGFR1- (pLeo1061), or EGFR-derived intracellular signaling domains (pLeo1165), and then cultivated in culture medium containing different concentrations of human cTnI. NanoLuc levels in culture supernatants were quantified at 24 h after cTnI stimulation. (C) The monoclonal CardioReport cell line for high-sensitivity detection of cTnI in vitro. CardioReport cells stably expressing TropR$_{FGFR2b}$, TetR-Elk1 and a TetR-driven NanoLuc expression unit were cultivated in culture medium containing different concentrations of cTnI. NanoLuc levels in culture supernatants were quantified after 24 h. (D) Minimal exposure time for activation of TropR-dependent gene expression. CardioReport cells were incubated in culture medium containing 100 ng/mL cTnI for different durations, after which the medium was exchanged to cTnI-free DMEM. NanoLuc levels in culture supernatants were quantified at 24 h after medium exchange. (E) Reversibility of cTnI-dependent target gene expression. CardioProtect cells were cultivated for 3 days while human cTnI levels in the culture medium were successively switched between 0 and 100 ng/mL. NanoLuc levels were measured every 8 h. The cell density was readjusted to $1 \times 10^5$ cells/ mL at every 24 h. (F) Doxycycline-based safety switch to reset TropR-mediated target gene expression. Doxycycline has the capability to hinder the binding and subsequent activation of TetR-Elk1 on TetR-specific promoters, regardless of the cell's MAPK activation status. Experimentally, CardioReport cells were cultivated in cell culture medium containing 100 ng/mL human cTnI and different concentrations of doxycycline (Dox). NanoLuc levels in culture supernatants were quantified after 48 h. Data in (B–F) are presented as mean ± SD, $n = 3$ individual experiments. Statistical significance was determined with Student's two-tailed $t$-test, ***($p < 0.001$), $p = 0.0005$ for Fig. 3D. Source data are available online for this figure.

onset (Kunadian and Gibson, 2012; Logallo et al, 2015). TNK is a 527-amino-acid recombinant tissue-type plasminogen activator that selectively converts thrombus-bound plasminogen to the serine protease plasmin, which in turn catalyzes degradation of the fibrin matrix of the thrombus (Kunadian and Gibson, 2012;

Logallo et al, 2015). Thus, coupling TropR-mediated cTnI detection (an early marker of AMI) to self-sufficient TNK secretion (a first-line thrombolytic drug) within a patient could contribute to minimizing the delay between symptomatic diagnosis of cardiac injury and therapeutic intervention. Here, we examined various

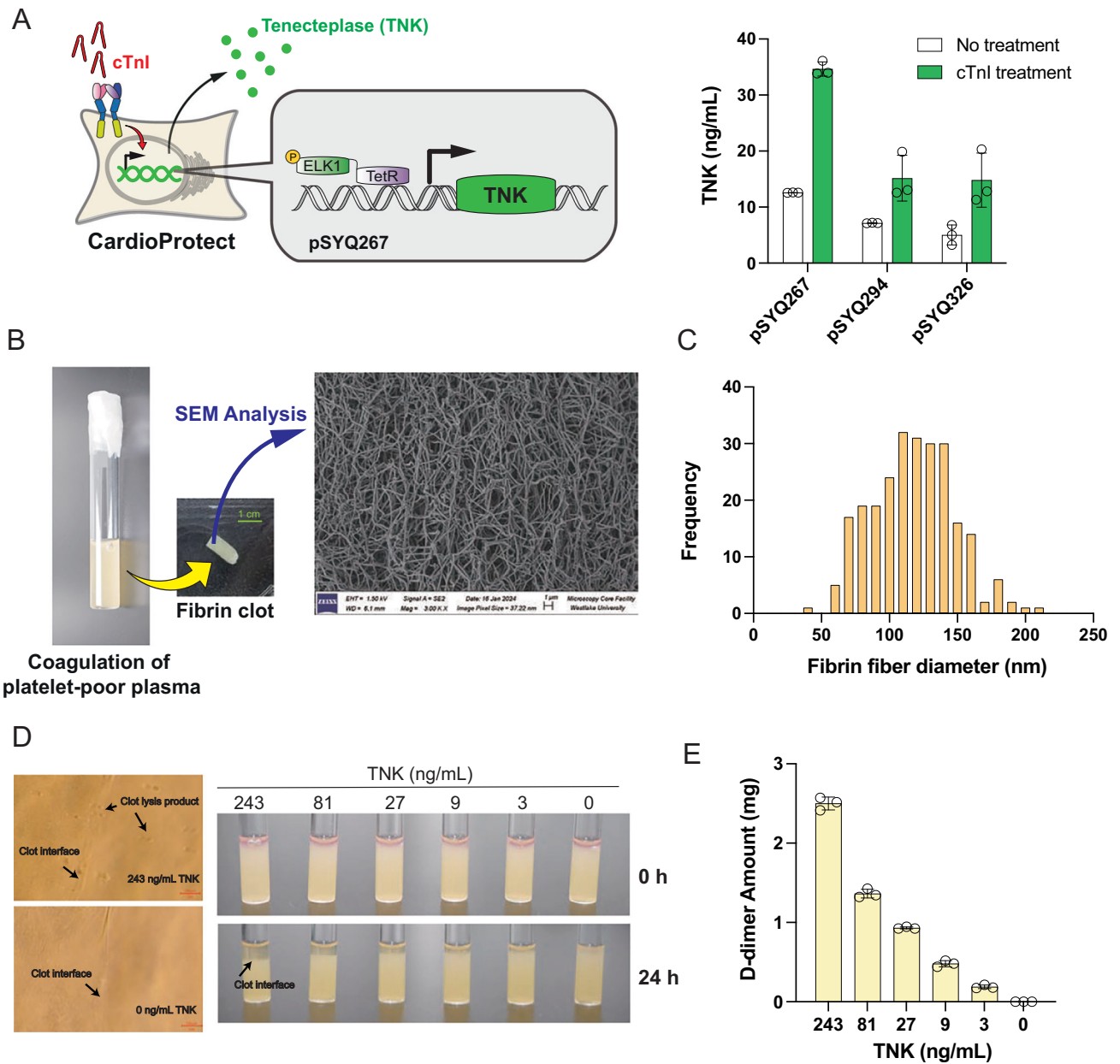

**Figure 4. Thrombolysis of fibrin clots by designer cell-produced tenecteplase (TNK).**

(**A**) Designer cells engineered for cTnI-triggered TNK secretion. HEK-293 cells were transfected with constitutive expression vectors for TropR$_{FGFR2b}$, TetR-Elk1 and different variants of TetR-driven TNK expression units (pSYQ267, P$_{hCMV*-1}$-TNK-pA; pSYQ294, P$_{hCMV*-1}$-NanoLuc-P2A-TNK-pA; pSYQ326, P$_{hCMV*-1}$-TNK-P2A-NanoLuc-pA), and incubated in cell culture medium containing 100 ng/mL cTnI. TNK levels in culture supernatants were quantified at 48 h post transfection by ELISA. Data were presented as mean ± SD, $n = 3$ individual experiments. (**B**) Morphological analysis of fibrin clots generated by coagulation of platelet-poor plasma in vitro. At 2 h after the addition of 10 mM CaCl$_2$ to platelet-poor plasma in a glass cuvette, gel-like fibrin clots were profiled (scale bars in green) and analyzed by scanning electron microscopy (SEM) at x3000 magnification. (**C**) Fiber size distribution of fibrin clots. Five 10000x images were obtained from the SEM analysis data shown in (**B**), and the diameters of 50 randomly picked fibers were estimated by ImageJ ($n = 250$ fibers). (**D**, **E**) Validation of dose-dependent fibrinolysis by mammalian cell-secreted TNK. Aliquots (20 μL) of conditioned medium of HEK-293 cells transfected with a constitutive TNK expression vector (P$_{hCMV}$-TNK-pA; pSYQ198) were serially diluted and added to freshly formed fibrin clots prepared as described above in (**B**). Microscopic analysis (**D**) and D-dimer amounts in the reaction system (**E**) were quantified after 24 h. Data in (**E**) are presented as mean ± SD of D-dimer levels of 2 replicate clot samples per group. Source data are available online for this figure.

expression systems for TNK secretion, and found that the TetR-driven monocistronic expression vector pSYQ267 (P$_{hCMV*-1}$-TNK-pA; P$_{hCMV*-1}$, tetO$_7$-P$_{hCMVmin}$) afforded the highest absolute expression levels and induction folds of cTnI-triggered TNK

production in TropR-transgenic cells, whereas bicistronic expression systems with TNK and NanoLuc placed into a same transcription unit appeared to show reduced overall TNK secretion capacity (Fig. 4A). Next, to quantify TNK efficacy, we established a

thrombolysis assay based on ex vivo plasma coagulation (Dhurat and Sukesh, 2014; Duffy et al, 2017). Specifically, plasma collected from human whole blood was mixed with 10 mM calcium chloride solution to generate fibrin clots, which showed morphologies (Fig. 4B) and fiber diameters (Fig. 4C) resembling those that typically form during the late stages of AMI or ST-segment elevation myocardial infarction (STEMI) in humans (Silvain et al, 2011; Rentrop, 2000; Siniarski et al, 2021). When applying TNK-containing conditioned media to this thrombolysis assay, we were able to validate TNK-specific clot lysis both through microscopic analysis (Fig. 4D) and dose-dependent generation of D-dimers, indicative of fibrin degradation (Strittmatter et al, 2022) (Fig. 4E).

Next, we generated monoclonal cell lines stably expressing the full genetic circuit for TropR-mediated TNK secretion using a non-viral random integration approach based on the Sleeping Beauty transposase technology (Kowarz et al, 2015), which typically yields an array of individual cell clones showing diverse profiles of relative fold-changes between maximal TNK production capacity and basal secretion levels (Xie et al, 2016; Wang et al, 2018) (Fig. 5A). As with any synthetic gene regulation system moving towards therapeutic applications, fundamental questions include whether and when any type of basal expression might translate into uncontrolled therapeutic effects, as well as estimation of the minimal efficacious dose of the therapeutic protein (Shao et al, 2024). To quantify the therapeutic efficacy of TNK, we tested different doses under the experimental conditions described for both prevention of clot formation in platelet-rich plasma (Appendix Fig. S4A) and for fibrinolysis of clots that were produced from platelet-poor plasma (Appendix Fig. S4B). Based on the findings of all related experiments (Fig. 4D; Appendix Fig. S4), we concluded that at least 3 ng/mL TNK was required to cause observable clot lysis, with saturating fibrinolytic effects starting to be recorded at 27 ng/mL TNK. At the same time, the in vitro system can tolerate a basal TNK secretion of up to 1 ng/mL, since no fibrinolytic effects were observed below that concentration (Appendix Fig. S4B). Based on these effective TNK concentrations as well as further experimental validation that the cells can be sufficiently stimulated by short "contact times" with cTnI in vivo (Appendix Fig. S5), we selected a clone that provided the most reasonable trade-off between low basal secretion and production of sufficient TNK levels (~200 ng/mL) during the fully activated state (Fig. 5B). Next, we encapsulated this monoclonal cell line, which we designated CardioProtect, in alginate-poly-L-lysine-alginate microspheres (Schukur et al, 2016) —a well-established immuno-isolating system for cell implantation therapies, showing excellent compatibility with human whole blood systems (Schukur et al, 2016). The cTnI-responsiveness of microencapsulated CardioProtect cells remained intact and dose-dependent (Fig. 5C); however, absolute levels of TNK secretion were compromised following the encapsulation process (Fig. 5B,C), likely due to the additional physical barrier created between the cells and their environment, required to isolate them from the host's immune cells for in vivo applications. Nevertheless, activated TNK production levels remained within the targeted efficacy window (>27 ng/mL; triggered by 100 ng/mL cTnI), with improved fold-changes being achieved either by pre-incubating CardioProtect in a doxycycline-containing medium before encapsulation or by treating encapsulated CardioProtect microbeads with 100 ng/mL doxycycline to reduce baseline expression (Fig. 5B). To illustrate its treatment potential, we established a culture model featuring (i)

generation of AMI-related fibrin clots with concomitant cTnI surges in human blood, (ii) closed-loop CardioProtect-mediated cTnI detection and TNK production, and (iii) self-sufficient cTnI-stimulated clot lysis within the same experimental system (Fig. 5D). Indeed, 100 ng/mL human cTnI triggered complete lysis of fibrin clots, whereas clot lysis was not observed in empty capsules not secreting TNK or with doxycycline-treated CardioProtect cells (Fig. 5E). This time-dependent clot lysis profile was consistent with the designed cTnI-inducible doxycycline-repressible TNK expression logic both in culture (Fig. 5C; Appendix Fig. S6) and in human blood (Fig. 5E,F; Appendix Fig. S7). Notably, even though baseline TNK levels produced by our selected CardioProtect cell clone (Fig. 5A) appeared to be insufficient to induce tonic signaling or produce stimulus-independent therapeutic effects such as premature clot lysis (Appendix Figs. S4 and S7), the doxycycline-dependent safety switch can effectively shut off both basal and fully-induced TNK production by encapsulated CardioProtect if required (Fig. 5E,F; Appendix Fig. S6). This result also suggests that cTnI-dependent activation of endogenous MAPK target genes only are incapable of inducing therapeutic effects in CardioProtect (Fig. 5E). Thus, this work demonstrates promising potential in meeting the molecular criteria necessary for the development of implantable biosensors aimed at detecting and treating acute myocardial infarction in humans.

## Discussion

In recent years, various cell therapy approaches based on native (Simpson et al, 2007) or genetically engineered mesenchymal stem cells (Zisa et al, 2009) and also CAR-T cells (Aghajanian et al, 2019; Rurik et al, 2022) have been developed with the aim of improving cardiac repair or attenuating cardiac fibrosis. However, these strategies target relatively late stages of cardiac injury and are therefore primarily applicable for management of patients who have survived the acute cardiac damage that accounts for most deaths. To significantly reduce mortality rates, treatments that enable rapid therapeutic intervention at the onset of acute myocardial infarction (AMI) are still needed (McDonnell et al, 2009). Although the overall mortality rate for AMI has declined over the past two decades in some industrialized countries due to increased utilization of PCI (Camacho et al, 2022; Dégano et al, 2015), prehospital cardiac arrest has emerged as a significant contributor to AMI mortality, presenting a challenge for which effective solutions are currently lacking (Toshima et al, 2021; Asaria et al, 2022). To address this issue, we aimed to engineer a cell-based system to sense surges of cardiac troponin I (cTnI) and respond with synchronized production and secretion of thrombolytic agents. For future clinical application, CardioProtect capable of sensing cTnI and producing TNK may have to be encapsulated in a semipermeable membrane such as alginate-poly-L-lysine-alginate (Xie et al, 2016; Wang et al, 2018) to provide a physical barrier that protects engineered cells from the host's immune response, while still providing access to the patient's bloodstream to allow flexible exchange of nutrients and metabolites across the capsule membrane, required for the engineered sense-and-response algorithm (Fig. 1A). Since this type of designer cell-based therapy physically segregates the encapsulated cell content from the host tissues, there would be only minor restrictions on the cellular background and/or

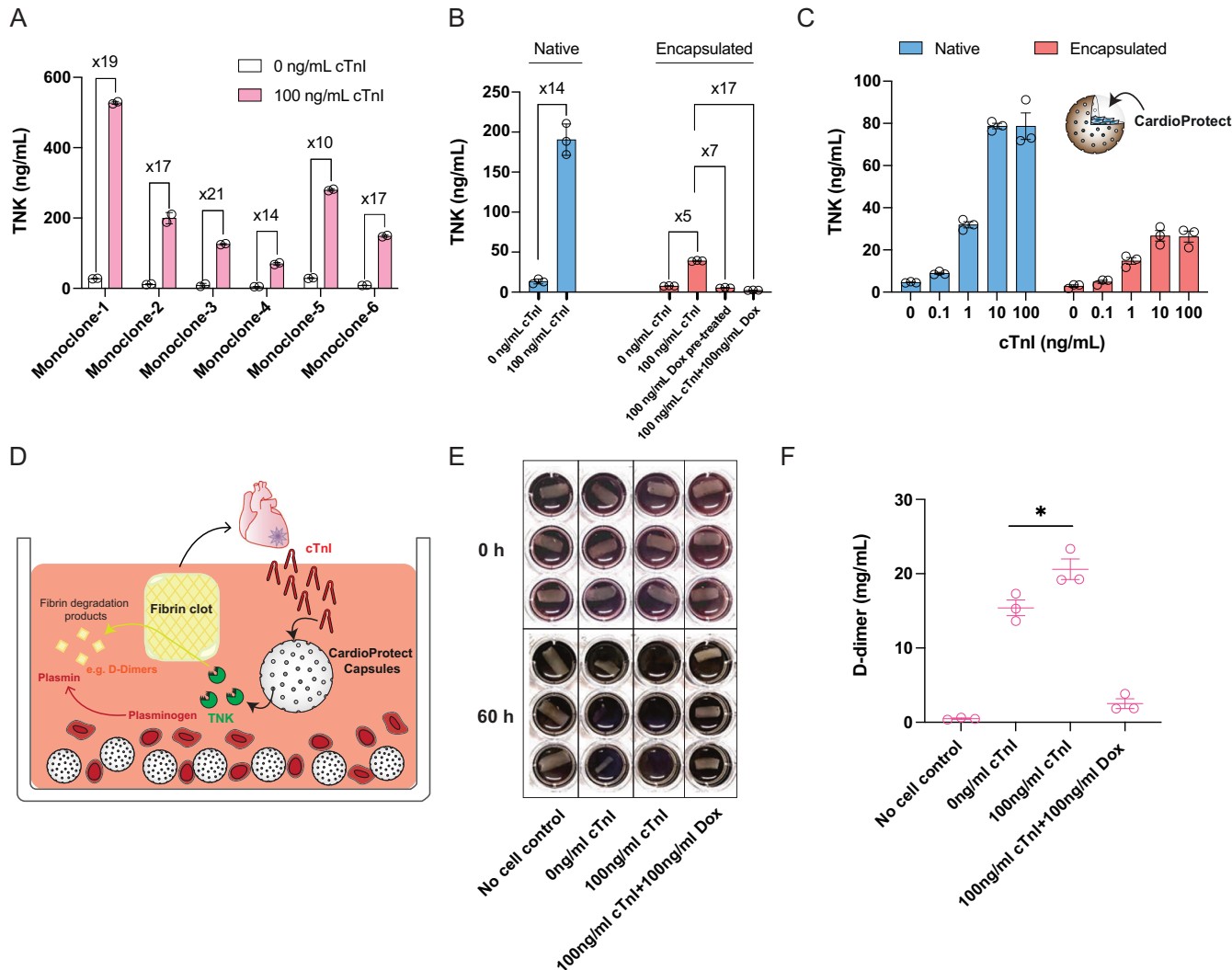

**Figure 5. Closed-loop control of cTnI-triggered thrombolysis in human blood culture.**

(A) Selection of monoclonal CardioProtect cell lines. Representative single cell clones stably expressing TropR$_{FGFR2b}$, TetR-Elk1, and a TetR-driven TNK expression unit (pSYQ367; ITR$_{SB}$-P$_{hCMV^*-1}$-TNK-pA-ITR$_{SB}$) were harvested and cultivated in medium containing 0 or 100 ng/mL human cTnI, and TNK levels in culture supernatants were quantified after 24 h. Data are presented as mean ± SD, $n = 2$ individual ELISA measurements. (B) cTnI-inducible doxycycline-repressible TNK production by monoclonal CardioProtect. One CardioProtect cell clone selected from (A) was harvested and encapsulated into alginate-poly-L-lysine-alginate microbeads with a density of 500 cells per capsule and a pore size of around 400 nm. About ~100 capsules (encapsulated) or 5 × 10⁴ CardioProtect cells (native) were seeded per well for cultivation in cTnI (0 or 100 ng/mL) and/or doxycycline-containing medium (100 ng/mL). Alternatively, cells were pre-treated with 100 ng/mL doxycycline prior to the encapsulation process. Secreted TNK levels in cell culture medium were quantified after 24 h by ELISA. (C) Dose-dependent TNK production by encapsulated CardioProtect. 6.5 × 10⁴ native or encapsulated CardioProtect cells were stimulated with different cTnI doses in culture and TNK levels in supernatants were quantified after 24 h by ELISA. Data were presented as the mean ± SEM; $n = 3$ independent experiments. (D) Potential mode of action of CardioProtect capsules during acute myocardial infarction (AMI). During AMI, fibrin clots (e.g., formed after rupture of atherosclerotic plaques) can cause cardiac injury and concomitant cTnI release into the bloodstream. Encapsulated CardioProtect cells designed to self-sufficiently detect systemic cTnI surges and release thrombolytic agents into the circulation should induce clot lysis and reduce further heart damage. (E, F) cTnI-dependent fibrinolysis by encapsulated CardioProtect cells. (E) In a 24-well plate, encapsulated CardioProtect cells were cultivated in human whole blood pre-mixed with an equal volume of RPMI 1640 medium, then a freshly prepared fibrin clot (prepared as described in Fig. 4B) was added together with inducer solutions of different concentrations (cTnI, 0 or 100 ng/mL; Dox, 0 or 100 ng/mL). Time-dependent clot lysis was monitored at the 60 h experimental endpoint. Co-culture of fibrin clots with empty capsules was included as negative control. (F) D-dimer levels in the blood system at 60 h post cTnI exposure. Data in (B) are presented as mean ± SD, $n = 3$ independent experiments. Data in (C, F) are presented as mean ± SEM, $n = 3$ independent experiments, statistical significance was determined with Student's two-tailed $t$-test, *($p < 0.05$), $p = 0.0409$ for Fig. 5F. Source data are available online for this figure.

transgene insertion strategies used to produce CardioProtect cells, as long as the engineered cells do not leak out of the capsules and enter the circulation. In this context, patients with a history of cardiac dysfunction and increased risk of AMI may be most likely

to benefit from a CardioProtect-based preventive therapy. Indeed, implanted CardioProtect may become a therapeutic solution of choice to detect early cardiac biomarkers (such as cTnI) that accumulate in the bloodstream and accordingly initiate therapeutic

interventions long before a patient would notice signs of discomfort.

To engineer CardioProtect, we focused on cTnI because it is a well-established serum biomarker reflecting the onset and severity of AMI in patients (Anderson and Morrow, 2017; McDonnell et al, 2009). Although various troponin assays exist for AMI diagnosis, no natural receptor has been identified, possibly because the physiological role of cTnI is confined to the regulation of intracellular processes in healthy cardiac tissue. Thus, to utilize systemic cTnI as a signaling molecule, we engineered a chimeric troponin receptor (TropR), capitalizing on a pair of cTnI-specific extracellular scFvs domains to trigger signal transduction pathways in the host cell upon ligand binding, thereby enabling troponin-regulated gene expression. Although other custom-designed antibody-receptor strategies rewiring antibody-ligand interactions to activation of intracellular signaling (e.g., chimeric antigen receptors (CAR) (Chakravarti and Wong, 2015)) or initiation of transgene expression (e.g., synthetic Notch receptor (synNotch) (Morsut et al, 2016)) have recently been described, CAR- and synNotch-based approaches are limited to sensing antigens located on the target cell surface (Scheller et al, 2018). TropR, in contrast, can be used to sense soluble target compounds (Scheller et al, 2018). Indeed, we confirmed that it can be used to detect cTnI in the clinically relevant range (1–100 ng/mL).

We therefore implemented a TropR-mediated cell therapy approach for the detection and treatment of cardiac injury by constructing CardioProtect cells to sense surges of cTnI and respond by producing and secreting TNK. This essentially recapitulates conventional emergency procedures while avoiding delays related to slow manifestation of physiological symptoms or logistic factors. To validate the design and functionality of CardioProtect cells, we next tested them in a custom-designed human whole blood-based reaction system. We adopted this approach because, in contrast to small animal models where experimental induction and development of myocardial infarction by coronary atherosclerosis is very challenging (Kumar et al, 2016; Golforoush et al, 2020), this human blood-based model incorporates most of the molecular and physiological features that would occur in AMI patients. First, because the fibrin clots generated in $CaCl_2$-treated human plasma are composed of 100% fibrin, have poor permeability for TNK, and closely resemble those formed through plaque rupture in AMI patients in terms of substantial fibrin composition (Silvain et al, 2011), we could evaluate the fibrinolysis efficacy of CardioProtect cells in strict conditions. Second, human whole blood contains the physiological factors required for the coagulation and fibrinolysis cascades, including the plasminogen pathway required for TNK action, in contrast to rodents, in which the components differ significantly (Gentry, 2004; Mohammed et al, 2020). Indeed, there is currently no mouse model that enables simultaneous induction of cardiac injury, cTnI elevation and assessment of thrombolytic action (Gao et al, 2010). We microencapsulated the CardioProtect cell in alginate-poly-L-lysine-alginate, since this is a proven, clinically available delivery system that would be compatible with further investigations of implantation therapies in vivo as soon as appropriate animal models are available (Schukur et al, 2016). Thus, the demonstration of closed-loop control of cTnI detection and initiation of TNK-dependent thrombolysis by encapsulated CardioProtect in human

blood served to validate the design and functionality of Cardio-Protect cells, providing a proof of concept, warranting further studies to explore systems with clinical utility, including the exploration of other cell types or delivery methods.

Next-generation cell therapies based on synthetic gene circuits can program therapeutic transgene expression in response to disease-specific or externally defined control signals (Mansouri and Fussenegger, 2022). However, closed-loop systems where secretion of protein therapeutics is triggered by increased endogenous disease metabolite levels may face safety and unpredictability concerns in a clinical setting (Lee et al, 2022). Thus, it is important to add an "open-loop" regulation interface that allows an external trigger compound to fine-tune, (re-)adjust and/or halt therapeutic transgene expression if necessary (Stefanov and Fussenegger, 2022). In CardioProtect cells, the closed-loop control circuit of cTnI-triggered thrombolytic release is under remote control of the FDA-approved drug doxycycline. Although currently approved TNK variants (e.g., TNKase, Genentech, US; Metalyse, Boehringer Ingelheim, EU) were developed specifically to minimize the risk of bleeding, such as the occurrence of systemic and intracranial hemorrhages (Warach et al, 2020), the option of doxycycline administration with CardioProtect cells provides an important safety layer, allowing interruption of the cell therapy and/or re-adjustment of the therapeutic dosage as required. As we observed a positive correlation between fibrinolytic activity and D-dimer levels in blood cultures, (hyper)activity of implanted CardioProtect in vivo may be evaluated by recording the plasma D-dimer levels, which may be a critical readout for a patient or doctor to decide whether either re-adjusting the dose or completely shutting down the system with the safety switch should be initiated. In contrast to most clinically tested safety switches, which are based on controlled expression of suicide genes (Di Stasi et al, 2011), the type of transcriptional-level gene switch used here does not kill the therapeutic cells but rather provides externally controllable "stop-and-go"-type gene expression with full tunability and reversibility. Therefore, we believe the present work may provide a basis for developing cell therapies for the self-sufficient early detection, treatment and prevention of cardiac injury.

## Methods

**Reagents and tools table**

| Reagent/resource | Reference or source | Identifier or catalog number |
|---|---|---|
| **Experimental models** | | |
| Human embryonic kidney cells (HEK-293T) | ATTC | CRL-11268 |
| Bone-marrow-derived immortalized mesenchymal stem/stromal cells (hMSC-hTERT) | (Simonsen et al, 2002) | |
| iCell Cardiomyocytes | FUJIFILM Cellular Dynamics | R1057 |
| C57/BL6J mice | Vital River Laboratories | |

| Reagent/resource | Reference or source | Identifier or catalog number |
|---|---|---|
| **Chemicals, enzymes and other reagents** | | |
| Phusion® High-Fidelity DNA Polymerase | New England Biolabs | Cat. No. M0530L |
| Restriction endonucleases | New England Biolabs | various |
| Antarctic Phosphatase | New England Biolabs | Cat. No. M0289L |
| T4 DNA Ligase | New England Biolabs | Cat. No. M0202L |
| Gibson Assembly® Master Mix | New England Biolabs | Cat. No. E2611S |
| XL10 gold competent cells | New England Biolabs | Cat. No. C2992 |
| ZR Plasmid Miniprep - Classic | Zymo Research | Cat. No. D4054 |
| ZymoPURE II Plasmid Midiprep Kit | Zymo Research | Cat. No. D4200 |
| Prostaglandin E1 (PGE1) | Merck Millipore | Cat. No. P5515 |
| Calcium chloride anhydrous ($CaCl_2$) | Sinopharm Chemical Reagent | Cat. No. 10005861 |
| Doxycycline hydrochloride | Sigma-Aldrich | Cat. No. D3447 |
| Reactive Red 120 (RR120) | Sigma-Aldrich | Cat. No. R0378 |
| Recombinant human cardiac troponin I (human cTnI) | Abcam | Cat. No. ab283299 Cat. No. ab207624 |
| Recombinant mouse cardiac troponin I (murine cTnI or TNNI3) | MyBioSource | Cat. No. MBS2012003 |
| Slow skeletal troponin I (TNNI1) | MyBioSource | Cat. No. MBS2012003 |
| L-Homoarginine hydrochloride | Sangon | Cat. No. A602842 |
| Magnesium chloride hexahydrate | Sangon | Cat. No. A610328 |
| Diethanolamine (DEA) | Macklin | Cat. No. D807525 |
| 4-Nitrophenyl phosphate disodium salt hexahydrate (pNPP) | Aladdin | Cat. No. 333338-18-4 |
| Recombinant human IL-6 | GenScript | Cat. No. Z03034 |
| Recombinant human insulin | Beyotime | Cat. No. P3376 |
| Dulbecco's modified Eagle's medium (DMEM) | Thermo Fisher Scientific | Cat. No. 10566016 |
| Fetal bovine serum (FBS) | Sigma-Aldrich | Cat. No. F7524 |
| RPMI 1640 medium | VivaCell | Cat. No. C3010 |
| Penicillin-streptomycin solution | Sigma-Aldrich | Cat. No. P4333 |
| Trypsin-EDTA | Thermo Fisher Scientific | Cat. No. 25300054 |

| Reagent/resource | Reference or source | Identifier or catalog number |
|---|---|---|
| PEI MAX | Polysciences | Cat. No. 24765 |
| ViaFect™ Transfection Reagent | Promega | Cat. No. E4981 |
| Opti-MEM™ I Reduced Serum Medium | Thermo Fisher Scientific | Cat. No. 31985062 |
| Puromycin dihydrochloride | Thermo Fisher Scientific | Cat. No. A1113803 |
| Blasticidin S HCl | Thermo Fisher Scientific | Cat. No. A1113903 |
| Zeocin | Thermo Fisher Scientific | Cat. No. R25005 |
| Hygromycin | Beyotime | Cat. No. ST1389 |
| Nano-Glo®Luciferase Assay System | Promega | Cat. No. N1120 |
| tPA Human ELISA kit | Thermo Fisher Scientific | Cat. No. BMS258-2 |
| Human D-dimer ELISA Kit | Elabscience | Cat. No. E-OSEL-H0010 |
| Human cTnI ELISA kit | Shanghai Enzyme-linked Biotechnology | Cat. No. ml023547 |
| **Software** | | |
| FlowJo™ (V.10.4) | https://www.flowjo.com/ | |
| ImageJ2 (V.2.14.0) | https://imagej.net/ | |
| GraphPad Prism (V.9) | https://www.graphpad.com/ | |
| **Other** | | |
| Infinite M1000 Multimode microplate reader | Tecan | |
| CytoFLEX LX flow cytometer | Beckman Coulter | |
| B-395 Pro Encapsulator | BÜCHI Labortechnik | |

## Vector design

References and construction details for all expression vectors are provided in Appendix Table S1. Sequences for related features are provided as a separate spreadsheet (Dataset EV1.xlsx).

## Cell culture and transfection

Human embryonic kidney cells and bone-marrow-derived immortalized mesenchymal stem/stromal cells were cultivated in Dulbecco's modified Eagle's medium supplemented with 10% fetal bovine serum (FBS) and 1% penicillin-streptomycin solution at 37 °C under a humidified atmosphere containing 5% $CO_2$. For passaging and cell culture, cells were detached by incubation in 0.05% trypsin-EDTA for 5 min at 37 °C, followed by the addition of 10 mL cell culture medium and centrifugation for 3 min at $200 \times g$ before reseeding the cells in new cell culture plates at a standard cell density ($1 \times 10^5$ cells/mL). Human induced pluripotent stem cell

(hiPSC) derived cardiomyocytes (iCell Cardiomyocytes) were plated and treated in accordance with the manufacturer's protocols. All these cell lines were authenticated and clear of mycoplasma contamination. HEK-293T, hMSC-hTERT and derivatives were transfected using home-made polyethyleneimine (PEI) solution (stock solution 1 mg/mL in ddH$_2$O) using a PEI:DNA ratio of 5:1 (w/w) and in a transfection volume of 50 µL serum-free DMEM per well. iCell Cardiomyocytes were transfected using ViaFect™ Transfection Reagent using a ViaFect:DNA ratio of 2:1 (v/w) and in a transfection volume of 50 µL serum-free Opti-MEM™ I Reduced Serum Medium per well. Unless indicated otherwise, transfection was performed 12 h after seeding $6.25 \times 10^4$ mammalian cells into each well of a 24-well plate. The cell culture medium was replaced with fresh medium not containing transfection reagents at 6 h after transfection.

## Flow cytometry

Transiently transfected cell populations were harvested through trypsinization and centrifugation, washed twice with ice-cold PBS, and analyzed with a CytoFLEX LX flow cytometer equipped for EGFP-detection (488 nm laser, 525/40 emission filter). 10,000 cells were recorded per dataset and analyzed with FlowJo™ software. The gating strategy involves initial FSC and SSC gates to isolate single, viable cells from the starting population, excluding debris and aggregates.

## Generation of stable cell lines

At 24 h after co-transfection of pCMV-T7-SB100 (Table S1) and corresponding SB-specific transposons in a 1:20 (w/w) ratio, the culture medium was replaced with fresh medium containing 3 µg/ mL puromycin dihydrochloride, 8 µg/mL blasticidin S HCl or 100 µg/mL zeocin selection reagent. After three rounds of cell passage in antibiotics-containing medium, surviving polyclonal cell populations were tested using corresponding analytical assays and/ or subjected to FACS-mediated single-cell cloning. Specifically, the monoclonal CardioReport cell line, transgenic for stable expression of Trop$_{FGFR2b}$, TetR-Elk1 and NanoLuc, was constructed by transfection of pBS804, pBS880, and pBS878 into HEK-293T cells, which were subsequently selected with corresponding antibiotics. Monoclonal CardioProtect cells, transgenic for stable expression of Trop$_{FGFR2b}$, TetR-Elk1 and TNK, was constructed by transfection of pSYQ367 into parental CardioReport cells followed by selection with 50 µg/mL hygromycin.

## Analytical assays

### SEAP assay

Expression levels of human placental secreted alkaline phosphatase (SEAP) in culture supernatants were quantified by means of a p-nitrophenylphosphate-based light absorbance time course assay. In brief, 80 µL of heat-inactivated supernatants (30 min at 65 °C) were transferred to a 96-well plate containing 100 µL of 2x SEAP assay buffer (20 mM L-homoarginine hydrochloride, 1 mM magnesium chloride, 21% diethanolamine, pH 9.8) per well. Immediately after the addition of 20 µL pNPP stock solution per well, the time-dependent increase of light absorbance at 405 nm was followed over 30 min using a Tecan Infinite M1000 Microplate Reader.

### NanoLuc assay

Nano luciferase levels were profiled using the Nano-Glo®Luciferase Assay System by adding 7.5 µL of a 50:1 (v/v) buffer-substrate mix to 7.5 µL sample per well in a black 384-well plate, followed by absorbance measurement with a Tecan Infinite M1000 Microplate Reader.

### TNK assay

Tenecteplase (TNK) levels in cell culture supernatants and animal serum were profiled using a tPA Human ELISA kit.

### D-Dimer assay

D-Dimer levels were profiled using a human D-dimer ELISA Kit.

### cTnI assay

Cardiac troponin I in animal serum was profiled using a human cTnI ELISA kit.

## Thrombolysis assay

Human blood samples from healthy donors were drawn under aseptic conditions using vacutainers containing 3.2% trisodium citrate at 9:1 v/v, supplemented with 0.1 µg/mL PGE1, and gently mixed by tube inversion before centrifugation at $330 \times g$ for 10 min. The supernatant was then collected as platelet-rich plasma (PRP), which was further centrifuged for 10 min (at 22 °C and $1000 \times g$) to collect platelet-poor plasma (PPP). Final platelet concentrations were about $1 \times 10^6$/mL as determined by flow cytometry. To induce clot formation, 10 mM CaCl$_2$ solution was added to 400 µL PPP in a Chrono-log cuvette, which was subsequently sealed with parafilm, gently inverted several times, and incubated at 37 °C for 2 h. To quantify clot lysis, 20 µL of tissue plasminogen activator (tPA) containing cell culture medium was added at 2 h after induction of clot formation. The time-dependent change of clot size was photo-recorded. Experiments involving human blood samples were performed according to the directives of the Ethics Committee of Westlake University (study number: 20240311XMQ001), with informed consent being obtained from all subjects. The experiments conformed to the principles set out in the WMA Declaration of Helsinki and the Department of Health and Human Services Belmont Report.

## Scanning electron microscopy

Fibrin clots were washed in 0.9 g/L saline for 2 h, then fixed in 2.5% glutaraldehyde solution. The next day, clots were washed in 0.1 M cacodylate buffer for 2 h on ice, and washed at least 3 times with ddH$_2$O before dehydration in a gradient of increasing ethanol concentrations (30, 50, 70, 90, and 100%). After dehydration in 100% ethanol for another $3 \times 10$-min cycles, clots were critical-point-dried, mounted onto stubs, and sputter-coated with gold. Clots were eventually imaged in five areas at two different magnifications (3000x and 10000x). Fiber diameter was measured by ImageJ based on five 10,000x images.

## Cell encapsulation

CardioProtect cells were encapsulated in coherent alginate-poly-(L-lysine)-alginate beads (400 µm; 500–1000 cells per capsule) using a

Büchi B-395 Pro Encapsulator set to the following parameters: a 200-μm nozzle with a vibration frequency of 1025 Hz, a 25-mL syringe operated at a flow rate of 410 units, and a voltage of 1.12 kV for bead dispersion. Capsules were further cultivated in standard cell culture medium or in human whole blood supplemented with 50% RPMI 1640 medium.

## Animal experiments

All experiments involving animals were performed according to the directive of the Institutional Animal Care and Use Committee of Westlake University, approved animal protocol number AP#23-052-XMQ. Eight-week-old healthy wild-type *C57/BL6J* male mice were purchased from Vital River Laboratories (Beijing, China) and housed in the Laboratory Animal Resources Center of Westlake University. Mice were kept at a constant temperature and humidity and exposed to 12 h cycles of alternating light and dark, with ad libitum access to standard rodent food and water. For analytical assays, blood was collected from the retroorbital plexus of mice by placing whole blood samples at room temperature for 1 h, followed by centrifugation for 10 min at $1500 \times g$ and 4 °C to obtain serum. All investigators were blinded to the treatment conditions.

## Data availability

The data that support the findings of this study are available on reasonable request to the corresponding authors. This study includes no data deposited in external repositories.

The source data of this paper are collected in the following database record: biostudies:S-SCDT-10_1038-S44320-025-00161-x.

## Peer review information

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

## Acknowledgements

We thank Hongjun Shi, Qidi Chen, Shuai Xue, Peng Bai, Pratik Saxena, Tobias Strittmatter, Elsa Görsch, and the Microscopy Core Facility of Westlake University for generous support and advice. Work in the laboratory of MX is supported by the Ministry of Science and Technology (MOST Project 2020YFA0909200), the National Natural Science Foundation of China (NSFC Project 32530063), the HRHI program 202209009 of the Westlake Laboratory of Life Sciences and Biomedicine, the Westlake Education Foundation and Tencent Foundation. Work in the laboratory of MF is supported by a European Research Council advanced grant (ElectroGene, no. 785800) and in part by the National Center of Competence in Research (NCCR) for Molecular Systems Engineering.

## Author contributions

**Yaqing Si**: Conceptualization; Validation; Investigation; Methodology; Writing—original draft; Writing—review and editing. **Yuxuan Fan**: Conceptualization; Validation; Investigation; Methodology; Writing—original draft; Writing—review and editing. **Leo Scheller**: Conceptualization; Validation; Investigation; Methodology; Writing—original draft; Writing—review and editing. **Bozhidar-Adrian Stefanov**: Conceptualization; Validation; Investigation; Methodology; Writing—original draft; Writing—review and editing. **Jian Lv**: Conceptualization; Validation; Investigation; Methodology; Writing—original

draft; Writing—review and editing. **Zhihua Wang**: Conceptualization; Validation; Investigation; Methodology; Writing—original draft; Writing—review and editing. **Mingqi Xie**: Conceptualization; Data curation; Formal analysis; Supervision; Funding acquisition; Writing—original draft; Project administration; Writing—review and editing. **Martin Fussenegger**: Conceptualization; Data curation; Formal analysis; Supervision; Funding acquisition; Writing—original draft; Project administration; Writing—review and editing.

Source data underlying figure panels in this paper may have individual authorship assigned. Where available, figure panel/source data authorship is listed in the following database record: biostudies:S-SCDT-10_1038-S44320-025-00161-x.

## Disclosure and competing interests statement

The authors declare no competing interests.

