## [Peer Review File · Molecular Systems Biology]

Engineering mammalian cells for detection and treatment of cardiac injury

Yaqing Si, Yuxuan Fan, Leo Scheller, Bozhidar-Adrian Stefanov, Jian Lv, Zhihua Wang, Mingqi Xie, and Martin Fussenegger

Corresponding author(s): Martin Fussenegger (fussenegger@bsse.ethz.ch) , Mingqi Xie (xiemingqi@westlake.edu.cn)

Review Timeline:

Submission Date:	2nd Oct 24
Editorial Decision:	29th Nov 24
Revision Received:	26th Jun 25
Editorial Decision:	25th Aug 25
Revision Received:	3rd Sep 25
Accepted:	26th Sep 25

Editor: Poonam Bheda

Transaction Report:

29th Nov 2024

Manuscript Number: MSB-2024-12665

Title: Engineering mammalian cells for detection and treatment of cardiac injury

Dear Dr. Fussenegger,

Thank you again for submitting your work to Molecular Systems Biology. We have now heard back from the three reviewers who agreed to evaluate your study. As you will see below, the reviewers appreciate that the proposed approach addresses a timely topic. However, they raise a series of concerns, which we would ask you to address in a major revision. Please note that Reviewer 2 has also included comments in an annotated version of your manuscript (see attachment).

Without repeating all the comments listed below, some of the more fundamental issues raised are the following:

- potential clinical/therapeutic implications are not clear
- timepoints and timecourses need justification and additional experimental support
- dose-response of microsphere encapsulated engineered cell line should be measured
- potential effects of endogenous pathway activation
- unclear whether further optimization of the receptor shows improvements

All other issues raised would need to be satisfactorily addressed. Please let me know in case you would like to discuss in further detail any of the comments, I would be happy to schedule a call.

We require:

- 1) A .docx formatted version of the manuscript text (including legends for main figures, EV figures and tables). Please make sure that the changes are highlighted to be clearly visible. Alternatively you may choose to submit your manuscript as a LaTeX file.
- 2) Individual production quality figure files as .eps, .tif, .jpg (one file per figure). For guidance, download the 'Figure Guide PDF' (<https://www.embopress.org/page/journal/17574684/authorguide#figureformat>).
- 3) At EMBO Press we ask authors to provide source data for the main figures. Our source data coordinator will contact you to discuss which figure panels we would need source data for and will also provide you with helpful tips on how to upload and organize the files.
- 4) A .docx formatted letter INCLUDING the reviewers' reports and your detailed point-by-point responses to their comments. As part of the EMBO Press transparent editorial process, the point-by-point response is part of the Peer Review File (PRF), which will be published alongside your paper.
- 5) A complete author checklist, which you can download from our author guidelines (<https://www.embopress.org/page/journal/17574684/authorguide#submissionofrevisions>). Please insert information in the checklist that is also reflected in the manuscript. The completed author checklist will also be part of the PRF.
- 6) Please note that all corresponding authors are required to supply an ORCID ID for their name upon submission of a revised manuscript.
- 7) It is mandatory to include a 'Data Availability' section after the Materials and Methods. Before submitting your revision, primary datasets produced in this study need to be deposited in an appropriate public database, and the accession numbers and database listed under 'Data Availability'. Please remember to provide a reviewer password if the datasets are not yet public (see <https://www.embopress.org/page/journal/17574684/authorguide#dataavailability>).

This study includes no data deposited in external repositories.

- 8) All Materials and Methods need to be described in the main text using our 'Structured Methods' format, which is required for all research articles. According to this format, the Methods section includes a Reagents and Tools Table (listing key reagents, experimental models, software and relevant equipment and including their sources and relevant identifiers) followed by a Methods and Protocols section describing the methods using a step-by-step protocol format. The aim is to facilitate adoption of

the methodologies across labs. Please upload the Reagents and Tools table as a separate document when submitting your revised manuscript. More information on how to adhere to this format as well as a downloadable template (.docx) for the Reagents and Tools Table can be found in our author guidelines:

<https://www.embopress.org/page/journal/17444292/authorguide#structuredmethods>

9) For data quantification: please specify the name of the statistical test used to generate error bars and P values, the number (n) of independent experiments (specify technical or biological replicates) underlying each data point and the test used to calculate p-values in each figure legend. The figure legends should contain a basic description of n, P and the test applied. Graphs must include a description of the bars and the error bars (s.d., s.e.m.). Please provide exact p values.

10) Our journal encourages inclusion of *data citations in the reference list* to directly cite datasets that were re-used and obtained from public databases. Data citations in the article text are distinct from normal bibliographical citations and should directly link to the database records from which the data can be accessed. In the main text, data citations are formatted as follows: "Data ref: Smith et al, 2001" or "Data ref: NCBI Sequence Read Archive PRJNA342805, 2017". In the Reference list, data citations must be labeled with "[DATASET]". A data reference must provide the database name, accession number/identifiers and a resolvable link to the landing page from which the data can be accessed at the end of the reference. Further instructions are available at .

11) We replaced Supplementary Information with Expanded View (EV) Figures and Tables that are collapsible/expandable online. A maximum of 5 EV Figures can be typeset. EV Figures should be cited as 'Figure EV1, Figure EV2" etc... in the text and their respective legends should be included in the main text after the legends of regular figures.

<https://www.embopress.org/page/journal/17574684/authorguide#expandedview>

13) Author contributions: CRedit has replaced the traditional author contributions section because it offers a systematic machine readable author contributions format that allows for more effective research assessment. Please remove the Authors Contributions from the manuscript and use the free text boxes beneath each contributing author's name in our system to add specific details on the author's contribution. More information is available in our guide to authors.

14) Disclosure statement and competing interests: We updated our journal's competing interests policy in January 2022 and request authors to consider both actual and perceived competing interests. Please review the policy

<https://www.embopress.org/competing-interests> and update your competing interests if necessary.

Share synopsis text and image, as well as eTOC:

Please note that these would be the final versions and changes during proofing are usually not allowed

16) As part of the EMBO Publications transparent editorial process initiative (see our policy here:

https://www.embopress.org/transparent-process#Review_Process), Molecular Systems Biology will publish online a Peer Review File (PRF) to accompany accepted manuscripts.

In the event of acceptance, this file will be published in conjunction with your paper and will include the anonymous referee reports, your point-by-point response and all pertinent correspondence relating to the manuscript. Let us know whether you agree with the publication of the PRF and as here, if you want to remove or not any figures from it prior to publication.

Please note that the Authors checklist will be published at the end of the PRF.

Molecular Systems Biology has a "scooping protection" policy, whereby similar findings that are published by others during review or revision are not a criterion for rejection. Should you decide to submit a revised version, I do ask that you get in touch after three months if you have not completed it, to update us on the status.

I look forward to receiving your revised manuscript.

Yours sincerely,

Poonam Bheda, PhD
Scientific Editor
Molecular Systems Biology

Reviewer #1:

Summary

The manuscript by Si et al. describes the development of a cellular circuit that can respond to physiological levels of an early marker of acute myocardial infarction (cTnI) by producing therapeutic levels of a thrombolytic protein (TNK). Preliminary studies were performed to optimise a cTnI-responsive receptor and signalling domain, using common reporter systems, before applying the system to regulate the expression of TNK. Alginate-encapsulated cells encoding this circuit were validated in vitro, and in the presence of human blood where the ability to lyse fibrin clots was assessed. The inclusion of a doxycycline-regulated transcription factor allows the circuit to be negatively-regulated providing a safety mechanism.

The authors have demonstrated that this receptor, which has been created using published scFv molecules and an established GEMMS receptor architecture, can indeed regulate expression of proteins of interest, including the therapeutic protein TNK. The advance is somewhat limited and technical in nature, providing researchers with a further example of the utility of the GEMMS approach. There was some optimisation performed, however, the system is not optimised such that it can be used clinically due to the leakiness observed. The interpretation and discussion of the results is lacking in several key areas, as is any forward-looking translational strategy. The study will be of interest to researchers studying synthetic receptor-based circuits and looking for alternative strategies to treat acute myocardial infarction.

Major points

1. Homodimeric vs heterodimeric receptor formats (Fig 1C) - there is no explanation of why the homodimeric format was superior with respect to fold-change induction. The contribution of expression level of the receptors (e.g. via FACS) needs to be ruled out to validate this finding. There is also no mention/explanation of the shift in the dose-response curves for the two formats.
2. Several figures, including 2B, 2C, 4A and 5F demonstrate that the circuit is leaky, yet this is not mentioned in the results, nor discussion. Please include this finding and provide an explanation and discuss the implications/how it might be mitigated.
3. The timecourse experiments in general were disappointing:
 - a. For the SEAP system (Fig 2C) the data was minimal and did not provide sufficient detail on the timings of the "off state" which is of importance for the translation of the system to patients. I strongly recommend a timecourse with more timepoints to ascertain exactly how long after removal of cTnI the circuit ceases to produce SEAP. Please also provide an explanation for the increase in SEAP expression over time for non-treated cells.
 - b. For the Nanoluc experiments (Fig 3F) it is unclear why this experiment was set up with a 24h readout after removal of media, especially considering the leakiness previously observed. As such, the conclusion that the circuit is activated after 6 minutes cannot be made. Rather, this experiment needs to be repeated whereby samples are taken and read out at the time specified.
4. The finding that encapsulated CardioProtect cells are less able to respond to cTnI/secrete TNK (Fig. 3B/C) is concerning, yet there is no explanation for/discussion around this. Importantly, a further experiment should be performed to reassess the dose-response of encapsulated cells to cTnI with respect to TNK production to demonstrate that these cells can function at, and provide physiological levels of cTnI and TNK respectively.
5. Please provide some insights into whether the in vitro blood-based system appropriately models the in vivo scenario in terms of blood flow and "contact time" between cTnI and the receptor. If it doesn't, can such a system be used to further validate the technology?
6. There is no indication of how this is anticipated to be used clinically, which I believe should be covered in the discussion e.g.
 - a. Where will the cells be implanted? If not the heart, do they need to be cardiomyocytes?

- b. This is a preventative medicine, so what patients will be given this treatment?
c. In terms of safety, what are the implications of several gene insertions in the genome? E.g. do the cells have the same phenotype as non-edited cells? Are they tumorigenic?

7. The argument for needing a safety switch was raised in the results section in the context of permanent elevation of STAT3 signalling but, seeing as the final construct is based on MAPK signalling, the impact of MAPK signalling in the CardioProtect cells also needs to be discussed.

8. Furthermore, do you understand the half-life of TNK in the blood and the safety implications of high and/or prolonged TNK in the blood? This cannot be mitigated using a safety switch seeing as the protein will have already been secreted. Please include this consideration in the discussion.

Minor points

Throughout: The results section is rather sparse on detail - please ensure that the reader can follow the main experimental points without having to refer to figure legends e.g. when describing transfection experiments please include what elements are being transfected.

p.2 - "synthetic-signalling-specific promoters" - this phrase is confusing and inaccurate. Some of the promoters used are endogenous and therefore not specific to the circuit e.g. STAT3 and NFAT, and the signalling molecules triggered are endogenous, rather than synthetic. Please re-word.

p. 5 - the first 10 lines of this results section re: TNK would fit better in the introduction where it is currently only included in the context of the circuit, with no prior mention.

p. 3 - the size limitations of delivery vehicles is mentioned here but there is no reference to the size of the hetero- and homodimerisable receptors to provide context for this. Please include.

p.3 - In reference to this section "Importantly, the linker between both scFv domains was carefully chosen to be short, so that the distance between the cTnl epitopes remains smaller than the distance between complementarity-determining regions (CDRs), thus preventing intrachain dimerization. In addition, these homodimeric receptors may show increased sensitivity by potentially binding two cTnl proteins simultaneously, thereby gaining in avidity and preventing nonfunctional receptor homodimers preformed within each receptor subunit". Unless I have misunderstood, the first sentence is referring to the distance between the cTnl epitopes on the 2 scFvs being smaller than the corresponding paratopes on the cTnl protein, not the CDRs (which belong to the scFv). If this is correct, then the importance of this is not to prevent intrachain dimerization (as this cannot occur) but to prevent non-functional homodimers where one cTnl molecule binds only to one receptor molecule and not 2. If this is the correct interpretation, please amend this section accordingly.

p.4 - Please provide clarity on what "those previously characterized" means in this sentence "cTnl-dependent activation of STAT3-regulated gene expression was achieved upon transient TropR transfection (Fig. 2A), showing comparable troponin sensitivity and fold-change to those previously characterized"

p.5 - Please remove the word "of" from this sentence "The VEGFR2int-containing TropR variant successfully activated of NFAT-specific promoters (Fig. 3C), but did not substantially differ from STAT3-based variants in terms of overall cTnl sensitivity, as shown in Fig. 1C and Fig. 2A"

p.5 - Please specify what cell background the CardioReport cells have been created with

p.5 - With reference to this sentence "Here, we examined various expression systems for TNK secretion, and found that the monocistronic expression vector pSYQ267 (PhCMV*-1-TNK-pA) afforded the highest absolute expression levels and induction folds of cTnl-triggered TNK production in TropR-transgenic cells" it is not clear in the text that these vectors are TetR-driven, nor why mono vs bicistronic versions were tested. Please include.

p.6 - Please specify in the results section how the TNK was provided (i.e. via conditioned media) in this experiment "Using this thrombolysis assay, we were able to validate TNK-specific clot lysis both through microscopic analysis (Fig. 4D) and dose-dependent generation of D-dimers, indicative of fibrin degradation"

Figure 3F - From the figure legend the use of the term "culture supernatants" implies that secreted protein is being measured, yet Nanoluc is an intracellularly expressed protein. Please clarify/correct the wording.

Figure 5A - please label the x axis with "clone" which is standard nomenclature, rather than "monoclone".

Figure 5B and C - please enlarge the text on the axes, which is difficult to read

Figure 5B - this figure needs to be combined with figure 5C as it is a control for the experiment in Fig 5C. In addition, please make it clear that when you state "The cTnl-responsiveness of microencapsulated CardioProtect cells remained intact; however, the efficiency of TNK secretion was compromised following the encapsulation process (Fig. 5C)" what comparison you are making (i.e the data in fig 5B vs the first 2 bars of Fig 5C).

p.6 - The phrase "Nevertheless, the secretion capacity, as indicated by fold-changes, could be restored either by pre-incubating CardioProtect in a doxycycline-containing medium before encapsulation or by treating encapsulated CardioProtect microbeads with 100 ng/mL doxycycline (Fig. 5C)" is misleading. I think what is meant here is that the secretion capacity could be reduced to basal levels, rather than "restored". Please amend.

Figure 5E - is each individual column in the plate a different condition? If so, please provide a label that can be read, and remove the empty wells from the figure. Please also provide the non-expert reader with a brief explanation of how to interpret the data in this figure, seeing as it is a qualitative measure.

Reviewer #3:

The manuscript by Yaqing Si and colleagues describes the development and validation of a new cellular therapy approach for ischemic heart diseases. The work is highly original, deploys state-of-the-art synthetic biology approaches (synthetic receptors), develops new sensitivity and input-output functions, applies it in relevant cell types, and tests it in new in vitro models of disease. I have a few comments on the approach overall, and on the specifics (see below), but overall the body of data is robust and merits strong consideration for publication.

High level comments:

- what is the therapeutic clinical use of the cells developed here? The two that come to my mind is either having these cells resident in a high-risk patient, or injecting an off-the-shelf product shortly after disease occurrence. These seem very futuristic, and far from any immediate applications. I think it would be helpful for the authors to comment on what they see as the future steps towards using a therapeutic cell product that uses the concept presented here in a clinical setting. For example: large animal studies? More realistic in vitro-systems? Or maybe the authors don't think this product has a viable path towards the clinic (which seems to be suggested by the fact that the authors do not seem to have filed IP protection around their approach), due to current technological limitations. I think it would be helpful for the reader to know what the authors are thinking around these topics.
- Some of the receptor constructs activate, at least in terms of design, endogenous cell pathways alongside the synthetic target gene. Is that true? I don't see experiments to test that in the current manuscript. Is that not relevant? Why not? Could some of the phenotypic outcomes attributed to synthetic gene activation be attributable to change in cell physiology downstream of activation of endogenous pathways? Either way is fine, would be good for the authors to guide the reader to interpret these nuances of the hybrid synthetic receptor design and applications.
- I could not get access to supplemental material (in case there was any), so I don't know if you publish your sequence of the plasmids/receptors. I think that should be a requirement. Ideally all the sequences, if not even just the relevant parts of the plasmids (promoter + coding sequence), or even just the receptor sequences. This would highly increase the impact and dissemination of the authors' work.
- some of the quantifications of the inductions of the receptor (e.g. in Fig. 2B,C), but maybe present also Fig.1 (although the dataset does not allow to evaluate), and again in fig. 4A, 5F, show a non-negligible basal activation. I could not find specific comments from the authors around this basal activation of the reporter downstream of the receptors. This raises questions for me: do the authors not notice? Do they notice but think it is not a problem at all? Do they notice, think it is a problem in the long term, and have ideas on how to address it? At this stage the technology is solid regardless of the answer, but for a publication I would expect the authors to at least provide their comments on this.
- The authors claim, with support from Fig. 2C mainly, that the receptor displays strict stop-and-go dynamics. I disagree that the presented dataset supports this claim. This is in part worsened by the fact that the basal keeps creeping up during the "stop" phase. See also comments in-text.
- Fig. 3 vs Fig. 1/2; in Fig. 3, the authors present a further optimization of their system compared to what presented in Fig. 1/2. Given that the optimization changes, to my understanding, concurrently both the receptor intracellular domain, reporter construct activated, and intermediate transcription factor, it makes it quite hard to compare Fig. 3 results with Fig. 1/2 results. Hence it makes it hard to claim that there was improvement compared to those early versions. To support those claims the authors would need to compare within the same at least reporter construct the different receptor variants.
- Fig. 5: given some of the difficulties in understanding what the authors did, it is somewhat hard to completely understand if the dataset supports the claims of the authors. See especially difficulty in understanding what the newly introduced assay is in Fig. 5D/E. It is really hard to understand the temporal evolution of the assay.

Specific points:

Abstract:

Would be good to specify at the abstract level which cell lines are you developing your constructs towards, at each stage, especially since you talk about single cell clones, monoclonal cell lines, etc; without giving any information of what cell lines are you engineering.

The 'human whole blood system' in the abstract sounds very mysterious.

- Fig. 3 E vs F: not super-clear why there is a delta of the NLUC readings (y axis in the 2 plots) between E and F, with E being lower by 10x approx? Maybe I'm missing something.
- Fig. 4A, right side: all the constructs show relatively high basal. Is that never a concern?
- Fig. 4B: 'PPP' in the figure, what does it mean?

- Fig. 4C: what do we compare these histograms to? Are they shown compared to treatment with known trombolitics or to therapeutic cells somewhere else?
- Fig. 4 A: quantification of TNK, how is it performed? Elisa? Would be good to specify in fig legend
- Fig. 5A: my note from first read was 'wow!' Really good display of receptor activation across multiple clones!!
- Fig. 5D: unfortunately from this plus the main text I could not get a really good picture of exactly what goes into this assay.
- Fig. 5E: I am not entirely sure what I am looking at here: I see many wells of a multi well plate, 12+12; are they the same wells at different time points (0 and 60h)? What do the different wells correspond to? There seem to be some writing on the lid of the plate, but from the figure I got it results illegible.
- Fig. 5F: there seems to be quite a bit of a basal, comparing condition 0ng/ml cTnI with the dox condition; is this not a problem clinically?

The rest of the specific points are in the word doc as comments in a file I emailed to the journal (through their online review system) on Nov 26 with the title:

MSB-2024-12665-Manuscript_Text-mstxt_Reviewer_Comments_Nov26
which represent an integral part of my review of this work.

MSB-2024-12665: Responses to the reviewers' comments (original comments are in blue)**Reviewer #1 (Comments to the Author):**

The manuscript by Si et al. describes the development of a cellular circuit that can respond to physiological levels of an early marker of acute myocardial infarction (cTnI) by producing therapeutic levels of a thrombolytic protein (TNK). Preliminary studies were performed to optimise a cTnI-responsive receptor and signalling domain, using common reporter systems, before applying the system to regulate the expression of TNK. Alginate-encapsulated cells encoding this circuit were validated in vitro, and in the presence of human blood where the ability to lyse fibrin clots was assessed. The inclusion of a doxycycline-regulated transcription factor allows the circuit to be negatively-regulated providing a safety mechanism.

The authors have demonstrated that this receptor, which has been created using published scFv molecules and an established GEMMS receptor architecture, can indeed regulate expression of proteins of interest, including the therapeutic protein TNK. The advance is somewhat limited and technical in nature, providing researchers with a further example of the utility of the GEMMS approach. There was some optimisation performed, however, the system is not optimised such that it can be used clinically due to the leakiness observed. The interpretation and discussion of the results is lacking in several key areas, as is any forward-looking translational strategy. The study will be of interest to researchers studying synthetic receptor-based circuits and looking for alternative strategies to treat acute myocardial infarction.

Throughout: The results section is rather sparse on detail - please ensure that the reader can follow the main experimental points without having to refer to figure legends e.g. when describing transfection experiments please include what elements are being transfected.

Major Comments:

1. Homodimeric vs heterodimeric receptor formats (Fig 1C) - there is no explanation of why the homodimeric format was superior with respect to fold-change induction. The contribution of expression level of the receptors (e.g. via FACS) needs to be ruled out to validate this finding. There is also no mention/explanation of the shift in the dose-response curves for the two formats.

To address this issue, we added EGFP-tags to both receptor configurations to enable flow cytometry-assisted comparison of TropR expression levels obtained from the transfected plasmid amounts used to produce the regulatory performances shown in **Fig. 1C** (200 ng of pYX943 encoding homodimeric TropR vs. 50 ng of pYX941 and 50 ng of pYX942 each encoding one subunit of heterodimeric TropR). The results show that the differences in cTnI-stimulated fold-change (Fig. 1C) were unrelated to differential TropR expression levels (**new Fig. S1**).

2. Several figures, including 2B, 2C, 4A and 5F demonstrate that the circuit is leaky, yet this is not mentioned in the results, nor discussion. Please include this finding and provide an

explanation and discuss the implications/how it might be mitigated.

Baseline expression is a common phenomenon with any kind of synthetic gene regulation system. One way to avoid/reduce baseline expression is the engineering and selection of stable monoclonal cell lines with satisfactory ON/OFF ratios (**Fig. 5A**). Another strategy is the integration of safety switches capable of shutting off either basal (**revised Fig. 5B**) or fully-induced transgene expression (**new Fig. S6**) using a suitable external control compound. In any case, a more fundamental question is whether such basal expression would translate into uncontrolled therapeutic effects in the context of biomedical applications. Here, following the advice of all reviewers (e.g., see point 4 of reviewer 3 below), we have added experiments showing that baseline TNK levels were insufficient to trigger thrombolysis in our experimental settings (compare clot morphologies and D-dimer levels of **new Fig. S4B** and **new Fig. S7**). Therefore, “tonic signaling”-dependent effects such as therapeutic activity potentially induced by basal expression are not only ruled out on an experimental basis (**new Fig. S7**), but also, if they were to occur, could be actively attenuated by the doxycycline-dependent safety switch by design (**revised Fig. 5B & new Fig. S6**). We have added these points in the revised manuscript.

3. The timecourse experiments in general were disappointing:

- a. For the SEAP system (Fig 2C) the data was minimal and did not provide sufficient detail on the timings of the "off state" which is of importance for the translation of the system to patients. I strongly recommend a timecourse with more timepoints to ascertain exactly how long after removal of cTnI the circuit ceases to produce SEAP.

Please also provide an explanation for the increase in SEAP expression over time for non-treated cells.

To demonstrate reversibility with “off state”-kinetics of the system, we chose an experimental setup described in “*Nat Biotechnol* (2010) 28(4):355” to study the time course of target gene expression following repeated cTnI stimulation (**new Fig. 3E**). The results show that once the cTnI trigger signal was removed by medium exchange, reporter gene expression rapidly became inactive and there was no accumulation beyond the “true” OFF-state level (i.e., expression level by cells never exposed to cTnI) within the next 24 h and until the next cTnI stimulation event (**new Fig. 3E**, red line). Based on these results, we do not anticipate that sustained baseline expression from non-treated cells would even approach the expression level of the ON-state, and thus stimulus-independent therapeutic effects are improbable, as has often been reported to be the case for other types of cell therapies (*J Immunol* (2019) 202(6):1735).

- b. For the Nanoluc experiments (Fig 3F) it is unclear why this experiment was set up with a 24h readout after removal of media, especially considering the leakiness previously observed. As such, the conclusion that the circuit is activated after 6 minutes cannot be made. Rather, this experiment needs to be repeated whereby samples are taken and read out at the time specified.

The reviewer was most likely referring to **Fig. 3D** (old Fig. 3E). Estimation of the minimal activation time by stimulation within fixed time-periods, followed by long-term cultivation in

trigger compound-free medium, is a common experimental setup in the field (*Science* (2016) 354(6317):1296; *Nat Biomed Eng* (2017) 1(1):5; *Nat Med* (2019) 25(8):1266). If a system was not sufficiently activated, cumulative reporter protein levels produced over 24 h would lie in a same range as for the 0 h exposure time point (i.e., cells that were never stimulated with cTnI; comparable to the steady OFF-state of **new Fig. 3E**). In fact, in this present case (**Fig. 3D**), given that the reporter protein level produced by stimulation for 0.1 h (6 min) significantly differs from that of the 0 h duration time point ($p = 0.0005$), we can conclude that exposure for only 6 min was sufficient to stimulate the cells. This was also the case when switching to TNK as the therapeutic output protein, where stimulation of CardioProtect for 6 min produced “higher” ON-states (**new Fig. S5B**, see also our response to point no. 5 below). We have added the statistical analysis data to **Fig. 3D**.

4. The finding that encapsulated CardioProtect cells are less able to respond to cTnI/secrete TNK (Fig. 3B/C) is concerning, yet there is no explanation for/discussion around this. Importantly, a further experiment should be performed to reassess the dose-response of encapsulated cells to cTnI with respect to TNK production to demonstrate that these cells can function at, and provide physiological levels of cTnI and TNK respectively.

This reviewer was most likely referring to **revised Fig. 5B** (which merges old Fig. 5B/5C). As suggested, we have added an experiment to show dose-dependent TNK production by either native or encapsulated CardioProtect at 24 h after cTnI stimulation. The data suggest that although cells indeed produce less TNK after encapsulation (in line with the previous results of Fig. 5B,5C), the overall sensitivity range of the system remains the same (i.e., significant response to 1 ng/mL cTnI and fully activated by 10 ng/mL cTnI), with encapsulated CardioProtect being capable of producing sufficiently high amounts of TNK to show clot lysis potential (above 3 ng/mL; **new Fig. S4**, see also our response to point 7 of reviewer 3 below). The main reason for the decreased total secretion amount of TNK is likely to be the additional physical barrier between the cells and its surroundings created by the alginate-based microcapsules, whose porosity was carefully chosen to exclude host immune cells, and could therefore impact protein diffusion (*J Microencapsul* (2002) 19(5):571). We have added some remarks on this issue to the revised manuscript.

5. Please provide some insights into whether the *in vitro* blood-based system appropriately models the *in vivo* scenario in terms of blood flow and "contact time" between cTnI and the receptor. If it doesn't, can such a system be used to further validate the technology?

To model an *in vivo* scenario mimicking drastic elevation dynamics of plasma cTnI, we added an experiment involving intravenous injection of a high cTnI dose (80 $\mu\text{g}/\text{kg}$) into mice. We then measured the decay profile of plasma cTnI over 6 h. The cTnI level was highest within the first 5 min, remained elevated for at least 30 min, and was apparently cleared from the circulation after more than 2 h (**new Fig. S5A**). These data suggest that there may be a “contact time” of approximately 30 min for cTnI to activate the receptor under these circumstances *in vivo*. Taking this together with the finding that our system requires a minimal contact time of 6 min between cTnI and the receptor for significant activation (**new Fig. S5B** and **Fig. 3D** (revised old Fig. 3E); also see our response to point 3b of reviewer 1 above), we may expect the system to operate

effectively in real AMI settings, where the decay of plasma cTnI levels could be even slower during prolonged heart damage (*Circulation* (2011) 124(21):2350).

6. There is no indication of how this is anticipated to be used clinically, which I believe should be covered in the discussion e.g. a. Where will the cells be implanted? If not the heart, do they need to be cardiomyocytes? b. This is a preventative medicine, so what patients will be given this treatment? c. In terms of safety, what are the implications of several gene insertions in the genome? E.g. do the cells have the same phenotype as non-edited cells? Are they tumorigenic?

We have integrated our responses to these important points into the revised manuscript. In brief:

a. yes, the cells will have to be implanted. Because the entire genetic machinery of CardioProtect (**Fig. 4A**) is fully synthetic and does not rely on cardiomyocyte-specific elements, and since the cells need to be encapsulated into a semipermeable membrane such as alginate-poly-L-lysine-alginate for implantation in any case, there is no restriction of host cell type as long as the encapsulated cells can connect to patient's bloodstream and mediate cTnI-sensing and TNK secretion in an "endocrine"-like fashion. We have added this perspective into a revised graphic illustration (**revised Fig. 1A**).

b. yes, patients with a history of cardiac dysfunction and increased risk of (sudden) emergence of myocardial infarction may most likely benefit from a CardioProtect-based preventive therapy (see also our response to point 1 of reviewer 3 below), as most AMI deaths occur within the first hour after the appearance of symptoms - often long before the patient is able to reach hospital. Therefore, implanted CardioProtect engineered to sense early cardiac biomarkers (such as cTnI) that accumulate in the bloodstream before the emergence of signs of discomfort and accordingly initiate therapeutic interventions (such as secretion of thrombolytics) can become instrumental in reducing AMI-related mortality.

c. The mode of action of a CardioProtect-based cell therapy is based on encapsulation of engineered cells, followed by implantation into a vascularized body site of the patient. Thus, there will be no genetic modification of the patient's own cells or tissue. Instead, only the encapsulated cells are genetically manipulated, and they are physically segregated from the patients' cells by the capsule membrane. Therefore, genetic modification would be only a minor safety issue. A greater risk factor for this type of cell therapy is potential leakage of the encapsulated cell contents into the circulation, which could in theory be tumorigenic. However, we used an encapsulation technique similar to what is currently in clinical use for the delivery of iPSC-derived beta-cells to type-1 diabetic patients (*Nat Med* (2016) 22(3):306; *Adv Healthc Mater* (2024) 13(19):e2400185), for which no such adverse issue has been reported to date.

7. The argument for needing a safety switch was raised in the results section in the context of permanent elevation of STAT3 signalling but, seeing as the final construct is based on MAPK signalling, the impact of MAPK signalling in the CardioProtect cells also needs to be discussed.

Please refer to our previous response. Because CardioProtect is encapsulated within a semipermeable membrane that physically segregates the engineered cells from the patient's own cells, the role of endogenous MAPK signaling within CardioProtect is expected to be of little

relevance as long as the engineered cells retain the ability to sense cTnI and produce TNK, and as long as the encapsulated cells do not leak out into the circulation. Nevertheless, we followed the advice of another reviewer and added an experiment to show that TropR (over)expression does not compromise the activation dynamics of corresponding endogenous pathways (see our response to point 2 of reviewer 3 below). Specifically, ectopic expression of MAPK-modulating Trop_{FGFR2b} did not impact on the cells' sensitivity to recombinant human insulin, which also triggers phosphorylation of ERK and Elk1 (*Nat Biomed Eng* (2017) 1(1):5) (**new Fig. S3B**), though it should be noted that to achieve insulin sensitivity through the same MAPK pathway, native insulin receptor had to be overexpressed for modulation of target gene expression. In the case of CardioProtect, where the cells are engineered to only (over)express TropR but not native insulin receptor, we would therefore expect neither compromise of endogenous signaling (e.g., by attenuating the physiological MAPK pathway) nor cross-sensitivity to TropR-unrelated ligands (e.g., by non-specifically responding to insulin) (**new Fig. S3B**). However, in the hypothetical case that unexpected activation does occur, we show that a safety switch is capable of shutting off transgene expression, either basal (**new Fig. 5B**) or fully induced (**new Fig. S6**).

8. Furthermore, do you understand the half-life of TNK in the blood and the safety implications of high and/or prolonged TNK in the blood? This cannot be mitigated using a safety switch seeing as the protein will have already been secreted. Please include this consideration in the discussion.

According to the literature, the half-life of tenecteplase is biphasic; the initial phase shows a mean half-life of 17-24 minutes and the mean terminal half-life was estimated to be in the range of 1-2 h (*Clin Pharmacokinet* (2002) 41(15):1229). According to our new experimental data, exogenous cTnI would be cleared from the bloodstream within 2 h (**new Fig. S5A**). Thus, taking this in combination with the results of our new reversibility experiment (**new Fig. 3E**), we may expect that TNK secretion quickly returns to basal levels once cTnI levels have decreased. Using the thrombolysis assay, we have also shown that this baseline expression (both the 0 ng/ml cTnI condition and dox condition) was incapable of triggering premature clot lysis (**new Fig. S7**). Nevertheless, even in the unexpected case that basal TNK secretion produces undesired side effects, we show that the doxycycline-based safety latch is able to fully attenuate transgene expression as well as excessive TNK production (**new Fig. S6**).

Minor points:

1. p.2 - "synthetic-signalling-specific promoters" - this phrase is confusing and inaccurate. Some of the promoters used are endogenous and therefore not specific to the circuit e.g. STAT3 and NFAT, and the signalling molecules triggered are endogenous, rather than synthetic. Please re-word.

We agree and have changed the term to "synthetic promoters containing response elements for endogenous signaling-specific transcription factors".

2. p. 5 - the first 10 lines of this results section re: TNK would fit better in the introduction

where it is currently only included in the context of the circuit, with no prior mention.

We have added background information on TNK to the revised introduction.

3. p.3 - the size limitations of delivery vehicles is mentioned here but there is no reference to the size of the hetero- and homodimerisable receptors to provide context for this. Please include.

We have added the requested information.

4. p.3 - In reference to this section "Importantly, the linker between both scFv domains was carefully chosen to be short, so that the distance between the cTnI epitopes remains smaller than the distance between complementarity-determining regions (CDRs), thus preventing intrachain dimerization. In addition, these homodimeric receptors may show increased sensitivity by potentially binding two cTnI proteins simultaneously, thereby gaining in avidity and preventing nonfunctional receptor homodimers preformed within each receptor subunit". Unless I have misunderstood, the first sentence is referring to the distance between the cTnI epitopes on the 2 scFvs being smaller than the corresponding paratopes on the cTnI protein, not the CDRs (which belong to the scFv). If this is correct, then the importance of this is not to prevent intrachain dimerization (as this cannot occur) but to prevent non-functional homodimers where one cTnI molecule binds only to one receptor molecule and not 2. If this is the correct interpretation, please amend this section accordingly.

We apologize for the misleading description, as similar concerns were also raised by another reviewer. Following discussion with experts in protein design, we decided to rephrase this sentence by removing all speculative statements about intrachain dimerization and non-functional homodimer formation. The revised paragraph therefore reads as follows: "Importantly, the linker between both scFv domains was carefully chosen to be short, so that the distance between the cTnI epitopes remains smaller than the distance between complementarity-determining regions (CDRs). In addition, these homodimeric receptors may show increased sensitivity by potentially binding two cTnI proteins simultaneously, thereby gaining in avidity".

5. p.4 - Please provide clarity on what "those previously characterized" means in this sentence "cTnI-dependent activation of STAT3-regulated gene expression was achieved upon transient TropR transfection (Fig. 2A), showing comparable troponin sensitivity and fold-change to those previously characterized"

We have clarified that we were referring to previous experiments that were carried out in HEK-293 cells.

6. p.5 - Please remove the word "of" from this sentence "The VEGFR2int-containing TropR variant successfully activated of NFAT-specific promoters (Fig. 3C), but did not substantially differ from STAT3-based variants in terms of overall cTnI sensitivity, as shown in Fig. 1C and Fig. 2A."

Corrected.

7. p.5 - Please specify what cell background the CardioReport cells have been created with

We have added the requested information in the revised manuscript.

8. p.5 - With reference to this sentence "Here, we examined various expression systems for TNK secretion, and found that the monocistronic expression vector pSYQ267 (PhCMV*-1-TNK-pA) afforded the highest absolute expression levels and induction folds of cTnI-triggered TNK production in TropR-transgenic cells" it is not clear in the text that these vectors are TetR-driven, nor why mono vs bicistronic versions were tested. Please include.

We have added the requested information to the description.

9. p.6 - Please specify in the results section how the TNK was provided (i.e. via conditioned media) in this experiment "Using this thrombolysis assay, we were able to validate TNK-specific clot lysis both through microscopic analysis (Fig. 4D) and dose-dependent generation of D-dimers, indicative of fibrin degradation (Fig. 4E)."

We have added this information to the revised manuscript.

10. Figure 3F - From the figure legend the use of the term "culture supernatants" implies that secreted protein is being measured, yet Nanoluc is an intracellularly expressed protein. Please clarify/correct the wording.

NanoLuc can be used in different forms: the unfused form of NLuc is indeed intracellular and provides high sensitivity and light output due to its intracellular stability, while a secreted form (containing an IgK-derived signal peptide at the N-terminus) is also available for applications requiring the enzyme to be secreted into the media (*Bioconjug Chem* (2016) 27(5):1175). We have consistently used the secreted form of NanoLuc throughout our work. We have added this information to the revised manuscript.

11. Figure 5A - please label the x axis with "clone" which is standard nomenclature, rather than "monoclonal".

We have fixed the labeling as requested.

12. Figure 5B and C - please enlarge the text on the axes, which is difficult to read

Following your request (together with your next point), we have combined these two figures and used a new, consistent font style (**revised Fig. 5B**).

13. Figure 5B - this figure needs to be combined with figure 5C as it is a control for the experiment in Fig 5C. In addition, please make it clear that when you state "The

cTnI-responsiveness of microencapsulated CardioProtect cells remained intact; however, the efficiency of TNK secretion was compromised following the encapsulation process (Fig. 5C)." what comparison you are making (i.e the data in fig 5B vs the first 2 bars of Fig 5C).

We agree and have combined both figures (**revised Fig. 5B**). As a result of the new dataset on dose-dependent TNK production by encapsulated CardioProtect (**new Fig. 5C**; see our response to major point no. 4 above), we believe that the comparison between "native cells" vs. "encapsulated cells" is now clear.

14. p.6 - The phrase "Nevertheless, the the secretion capacity, as indicated by fold-changes, could be restored either by pre-incubating CardioProtect in a doxycycline-containing medium before encapsulation or by treating encapsulated CardioProtect microbeads with 100 ng/mL doxycycline (Fig. 5C)." is misleading. I think what is meant here is that the secretion capacity could be reduced to basal levels, rather than "restored". Please amend.

We agree and have rephrased the sentence accordingly.

15. Figure 5E - is each individual column in the plate a different condition? If so, please provide a label that can be read, and remove the empty wells from the figure. Please also provide the non-expert reader with a brief explanation of how to interpret the data in this figure, seeing as it is a qualitative measure.

We agree and have improved the presentation. Following the critique of all reviewers, we have replaced the handwritten labels with electronic fonts and grouped all replicates according to the same experimental condition.

Reviewer #3 (Comments to the Author):

The manuscript by Yaqing Si and colleagues describes the development and validation of a new cellular therapy approach for ischemic heart diseases. The work is highly original, deploys state-of-the art synthetic biology approaches (synthetic receptors), develops new sensitivity and input-output functions, applies it in relevant cell types, and test it in new in vitro models of disease.

I have a few comments on the approach overall, and on the specifics (see below), but overall the body of data is robust and merits strong consideration for publication.

We greatly appreciate your positive evaluation and many helpful comments. In addition to your major and minor comments, to which we respond below, you inserted a large number of additional comments directly in the text file, and we were not sure how best to respond to them. Finally, rather than inserting our responses as further comments in the text, we decided to submit a "clean" revised manuscript file according to common editorial practice and respond to all your comments by moving them to the end of this document. Therefore, our responses to your additional in-text comments are included here after our responses to your major and minor comments, under the

heading “Responses to your additional in-text comments”. Corresponding changes in the text are shown in red font, in the same way as other changes. We hope this is satisfactory.

Major Comments:

1. What is the therapeutic clinical use of the cells developed here? The two that come to my mind is either having these cells resident in a high-risk patient, or injecting an off-the-shelf product shortly after disease occurrence. These seem very futuristic, and far from any immediate applications. I think it would be helpful for the authors to comment on what they see as the future steps towards using a therapeutic cell product that uses the concept presented here in a clinical setting. For example: large animal studies? More realistic in vitro-systems? Or maybe the authors don't think this product has a viable path towards the clinic (which seems to be suggested by the fact that the authors do not seem to have filed IP protection around their approach), due to current technological limitations. I think it would be helpful for the reader to know what the authors are thinking around these topics.

This important question was also raised by another reviewer (please see our response to point 6 of reviewer 1 above). Yes, a possible administration route for this type of cell therapy would first require encapsulation of the engineered CardioProtect cells into an immune-isolating semipermeable membrane that avoids physical contact between the engineered cells and the patient's own tissues to increase safety, followed by implantation into a vascularized body site to enable flexible exchange of nutrients and metabolites so the engineered cells can interact with the host bloodstream exclusively by sensing disease markers (e.g. cTnI) and producing therapeutic agents (e.g. TNK). As correctly pointed out by this reviewer, patients with a history of cardiac dysfunction and increased risk of sudden (re)-emergence of myocardial infarction may most likely benefit from such a CardioProtect-based preventive therapy, as most AMI deaths occur within the first hour after the appearance of symptoms and often long before the patient is able to reach hospital. Therefore, this group of patients could receive CardioProtect-containing implants designed to sense cardiac injury and initiate therapeutic interventions at the earliest possible time (i.e., even before the patient feels specific signs of discomfort). And indeed, a major limitation of this project was the unavailability of a small animal model allowing for experimental induction and development of myocardial infarction by coronary atherosclerosis. The gold-standard mouse model for myocardial infarction is based on permanent ligation of the coronary artery (*Dis Model Mech* (2020) 13(11):dmm046565), which may produce heart damage (and also a cTnI increase), but this type of physical manipulation cannot be technically treated by thrombolytic drugs. Therefore, this would not be a suitable model of the “real-life” medical scenario of AMI patients, for which our CardioProtect therapy was initially designed. Further, larger animals such as pigs, sheep and dogs are more appropriate models for acute myocardial infarction from the viewpoint of translational medicine (*Br J Pharmacol* (2022) 179(5):770), but are suitable facilities are unfortunately not available to us. Therefore, we decided to focus on a custom-designed human whole-blood-based reaction system in which the experimental conditions are set in such a way that most of the molecular and physiological features occurring in AMI patients may reasonably be taken into account (as agreed by this reviewer in the in-text comment: “Very strong paragraph with very sensible explanation of support to your ex-vivo model”). Nevertheless, we are of course

aware that the current status of this technology remains far from translation to clinical practice, and this is also the reason why we are not pursuing strong IP-protection/commercialization plans at present.

2. Some of the receptor constructs activate, at least in terms of design, endogenous cell pathways alongside the synthetic target gene. Is that true? I don't see experiments to test that in the current manuscript. Is that not relevant? Why not? Could some of the phenotypic outcomes attributed to synthetic gene activation be attributable to change in cell physiology downstream of activation of endogenous pathways? Either way is fine, would be good for the authors to guide the reader to interpret these nuances of the hybrid synthetic receptor design and applications.

This is an excellent point. Indeed, all the TropR-dependent gene expression systems developed in this study utilize endogenous cell pathways, such as JAK/STAT3 and MAPK/ERK. To consider potential crosstalk, we have added experiments which show that ectopic expression of various TropR constructs does not compromise the activation dynamics of corresponding endogenous pathways. Specifically, overexpression of STAT3-modulating TropR_{IL6} did not impact the cells' sensitivity to IL6, which also signals through the STAT3 axis (**new Fig. S3A**). Likewise, overexpression of MAPK-modulating TropR_{FGFR2b} did not impact the cells' sensitivity to recombinant human insulin, which also triggers phosphorylation of ERK and Elk1 (*Nat Biomed Eng* (2017) 1(1):5) (**new Fig. S3B**). Note that co-expression of "native receptors" for IL6 (**new Fig. S3A**) and insulin (**new Fig. S3B**) was necessary to modulate target gene expression in either case, suggesting little interference with endogenous pathways in the present case, where the cells are engineered to (over)express only TropR but no native receptor to endogenous ligands (such as IL-6R or insulin receptor). Thus, we do not anticipate any compromise of endogenous signaling (e.g., by attenuating the physiological MAPK pathway) or any cross-sensitivity to TropR-unrelated ligands (e.g., by responding to insulin or IL-6).

3. I could not get access to supplemental material (in case there was any), so I don't know if you publish your sequence of the plasmids/receptors. I think that should be a requirement. Ideally all the sequences, if not even just the relevant parts of the plasmids (promoter + coding sequence), or even just the receptor sequences. This would highly increase the impact and dissemination of the authors' work.

We have provided all relevant sequences in our revised manuscript by uploading a separate spreadsheet.

4. Some of the quantifications of the inductions of the receptor (e.g. in Fig. 2B, 2C), but maybe present also Fig.1 (although the dataset does not allow to evaluate), and again in 4A and 5F, show a non-negligible basal activation. I could not find specific comments from the authors around this basal activation of the reporter downstream of the receptors. This raises questions for me: do the author not notice? Do they notice but think it is not a problem at all? Do they notice, think it is a problem in the long term, and have ideas on how to address it? At this stage the technology is solid regardless of the answer, but for a publication I would expect the

authors to at least provide their comments on this.

Please refer to our response to point 2 of reviewer 1 above. Baseline activation of gene expression is a common phenomenon with any kind of synthetic regulation system. One way to avoid/reduce baseline expression is the engineering and selection of stable monoclonal cell lines that show satisfactory ON/OFF ratios (**revised Fig. 5A**). Another strategy is the integration of safety switches capable of shutting off either basal (**revised Fig. 5B**) or fully-induced transgene expression using external control compounds of interest (**new Fig. S6**). In any case, a more fundamental question is whether such basal expression would translate into uncontrolled therapeutic effects in the context of biomedical applications. To address this, we have added experiments showing that baseline TNK levels were insufficient to trigger thrombolysis in our experimental settings (compare clot morphologies and D-dimer levels of **new Fig. S4B** and **new Fig. S7**). Therefore, tonic signaling-dependent effects such as therapeutic effects potentially induced by basal expression are not only ruled out on an experimental basis (**new Fig. S7**), but also, if they occurred, could be actively attenuated by the doxycycline-dependent safety switch by design (**revised Fig. 5B, new Fig. S6**). We have added these points to the revised manuscript.

5. The authors claim, with support from Fig. 2C mainly, that the receptor displays strict stop-and-go dynamics. I disagree that the presented dataset support this claim. This is in part worsened by the fact that the basal keeps creeping up during the "stop" phase. See also comments in-text.

We agree and have removed the "stop-and-go" term. To better describe the dynamics of receptor-mediated gene expression, we have added a reversibility experiment showing strict cTnI-dependent transcriptional activation (**new Fig. 3E**; see also our response to point 3a of reviewer 1 above). The results show that when the cTnI trigger signal was removed by medium exchange, reporter gene expression rapidly became inactive and there was no significant accumulation to produce "leaky expression" within the next 24 h and until the next cTnI stimulation event (**new Fig. 3E**, red line). These results indicate that basal activity would not lead to accumulation of expressed levels that are anywhere near the "real" ON-state level, and so there should be little likelihood of stimulus-independent therapeutic effects such as tonic signaling (*J Immunol* (2019) 202(6):1735).

6. Fig. 3 vs Fig. 1/2; in Fig. 3, the authors present a further optimization of their system compared to what presented in Fig. 1/2. Given that the optimization changes, to my understanding, concurrently both the receptor intracellular domain, reporter construct activated, and intermediate transcription factor, it makes it quite hard to compare Fig. 3 results with Fig. 1/2 results. Hence it make it hard to claim that there was improvement compared to those early versions. To support those claims the authors would need to compare within the same at least reporter construct the different receptor variants.

We are grateful for this critical and important remark. For better consistency, we have performed the experiment of Fig. 3B and Fig. 3C using SEAP as the reporter (i.e. the same reporter construct used for Fig. 1/2). The results show that the systems behave similarly across all settings with the

optimization principles for receptor domains remaining consistent across various isogenic configurations (**new Fig. S2B**).

7. Fig. 5: given some of the difficulties in understanding what the authors did, it is somewhat hard to completely understand if the dataset supports the claims of the authors. See especially difficulty in understanding what the newly introduced assay is in Fig. 5D/E. It is really hard to understand the temporal evolution of the assay.

To address this point, we studied the temporal evolution of the clot lysis process illustrated in Fig. 5 (**new Fig. S7**). As long as TNK was produced in sufficient amounts (e.g., triggered by at least 10 ng/mL cTnI), complete clot lysis can be observed after 36 h. Therefore, the 60 h data previously shown in Fig. 5E,5F represent endpoint measurements of this time-dependent clot lysis process, showing that there was no fibrinolytic activity in the basal 0 ng/mL cTnI condition or in the dox condition. These results support insignificant tonic signaling and full functionality of the safety switch over the entire timecourse (**new Fig. S7**).

Minor points:

1. Would be good to specify at the abstract level which cell lines are you developing your constructs towards, at each stage, especially since you talk about single cell clones, monoclonal cell lines, etc; without giving any information of what cell lines are you engineering.

The 'human whole blood system' in the abstract sounds very mysterious.

We have added the requested information into the abstract. We also changed 'human whole blood system' to "ex vivo blood culture system" in the abstract.

2. Fig. 3E vs F: not super-clear why there is a delta of the NLUc readings (y axis in the 2 plots) between E and F, with E being lower by 10x approx? Maybe I'm missing something.

We have checked the experimental setup and note that cells in **Fig. 3F** were exposed to 100 ng/mL cTnI and doxycycline for 48 h (and not 24 h). This explains why NanoLuc levels in Fig. 3F were substantially higher than in **Fig. 3D** (revised old Fig. 3E), where cells were only exposed to cTnI for 24 h. We have corrected the corresponding figure legends.

3. Fig. 4A, right side: all the constructs show relatively high basal. Is that never a concern?

This is a common feature in creating and optimizing synthetic gene regulation systems. Please refer to our responses to point 2 of reviewer 1 and your major point no. 4 above. While Fig. 4A was produced by transient transfection of plasmids encoding for Trop_{FGFR2b}, TetR-Elk1 and TetR-driven TNK-expression units, which resulted in high basal expression, Fig. 5A shows how individual cell clones with variable ON/OFF profiles and reduced basal expression can be carefully selected after stable integration of the genetic componentry using the Sleeping Beauty

transposase system (*Science* (2016) 354(6317):1296; *Nat Biomed Eng* (2018) 2(2):114). Because therapeutic applications are eventually built on stable cell clones with known and extensively characterized profiles (**revised Fig. 5A**, **revised Fig. 5B**), whether or not a “precursor” system created by transient transfection shows high basal expression may ultimately be irrelevant from a therapeutic point of view.

4. Fig. 4B: 'PPP' in the figure, what does it mean?

We have now added the full name (platelet-poor plasma) to the display item.

5. Fig. 4C: what do we compare these histograms to? Are they shown compared to treatment with known trombolitics or to therapeutic cells somewhere else?

The structure and properties of these fibrin fibers, including their diameter, define the mechanical properties of the clots, such as elasticity and resistance to thrombolytics. The clots formed in AMI patients typically show relatively thin fibers, where the degradation is significantly reduced with respect to healthy controls (*Arterioscler Thromb Vasc Biol* (2014) 34(7):1355). To test potential fibrinolysis effects of CardioProtect in human-relevant settings, we prepared clots with 100% fibrin and similar distributions of fiber diameters to those described for AMI patients.

6. Fig. 4A: quantification of TNK, how is it performed? Elisa? Would be good to specify in fig legend

Yes, TNK was measured by ELISA. We added this information to the revised figure legends.

7. - Fig. 5A: my note from first read was 'wow!' Really good display of receptor activation across multiple clones!!

- Fig. 5D: unfortunately from this plus the main text I could not get a really good picture of exactly what goes into this assay.

- Fig. 5E: I am not entirely sure what I am looking at here: I see many wells of a multi well plate, 12+12; are they the same wells at different time points (0 and 60h)? What do the different wells correspond to? There seem to be some writing on the lid of the plate, but from the figure I got it results illegible.

Fig. 5A shows a panel of monoclonal cell lines created by stable transfection of plasmids encoding for Trop_{FGFR2b}, TetR-Elk1 and TetR-driven TNK-expression units - the full genetic componentry required to produce CardioProtect cells (Fig. 4A). One of these clones was picked for expansion and encapsulation into alginate-poly-L-lysine-alginate microbeads (revised Fig. 5B), which may either be used for implantation therapies in humans (see our response to point 1 of reviewer 3 above), or for any *ex vivo* assay that involves interplay with human blood (**Fig. 5D**). As we outlined above (major point no. 1 of this reviewer), we chose the latter option because animal models capable of capitulating all clinically relevant AMI scenarios were out of reach. Thus, we set up the blood culture model in such a way that a) human blood, b) fibrin clots and c) encapsulated “therapeutic” cells were all placed into the same reaction system, so thrombolytic

effects produced by cTnI-mediated TNK release can be effectively visualized in terms of the dissolution profile of fibrin clots (**Fig. 5E,5F**). While Fig. 4B,4C confirm that the fibrin clots generated using our method indeed have similar morphologies, composition and size to those formed after rupture of atherosclerotic plaques in humans (see also our response to your minor point no. 5 above), Fig. 4D,4E showed that TNK produced by our CardioProtect cells indeed has the capacity to lyse these clots. Now, following the request of all reviewers, we have also characterized the efficacy window by showing that secreted TNK can not only trigger fibrinolysis of clots produced from platelet-poor-plasma (**new Fig. S4B**), but also prevent clot formation in platelet-rich plasma (**new Fig. S4A**). Furthermore, we found that at least 1-3 ng/mL TNK seems to be required to observe significant clot lysis throughout all assays (**new Fig. S4**). At the same time, saturating fibrinolytic effects were observed with 27 ng/mL TNK, indicating that encapsulated CardioProtect can produce sufficient TNK amounts in its fully activated state (40 ng/mL; **revised Fig. 5B**), even though non-encapsulated cells had greater production rates (200 ng/mL; **revised Fig. 5B**). Thus, the reduced TNK secretion capacity after the encapsulation process seems not to be a big issue. Furthermore, following your request to add a temporal evolution profile for the entire thrombolysis process (see our response to your major point no.7 above), we show that complete clot lysis is usually achieved at 36 h after exposure to TNK (**new Fig. S7**). Thus, the 60 h data shown in **Fig. 5E,5F** represent endpoint measurements of this process, showing that there was no fibrinolytic activity at basal 0 ng/mL cTnI or in the dox condition and indicating insignificant tonic signaling and full functionality of the safety switch over the entire time course (**new Fig. S7**). Also, to improve the data presentation, we have changed the handwritten labels in **Fig. 5E,5F** to electronic fonts and grouped all replicates for the same experimental condition (see our response to minor point no. 15 of reviewer 1 above).

8. Fig. 5F: there seems to be quite a bit of a basal, comparing condition 0ng/ml cTnI with the dox condition; is this not a problem clinically?

Following your request to show a temporal evolution of the entire thrombolysis process (see our response to your major point no. 7 above), we have repeated this experiment and show that there was no fibrinolytic activity in the basal 0 ng/mL cTnI condition or the dox condition (**new Fig. S7**). This conclusion was corroborated by the correlations among effective TNK concentration, endpoint clot morphology and D-dimer levels in the reaction system (**new Fig. S4**). Nevertheless, even if this basal TNK secretion unexpectedly produced undesired side effects, the doxycycline-based safety latch would be able to fully turn off gene expression irrespective of whether TNK production was previously fully active (**new Fig. S6**) or only basal (**new Fig. S7**).

Responses to your additional in-text comments:

As mentioned initially, we will respond to your additional in-text comments here. Corresponding changes in the text are shown in red font, in the same way as other changes. Please note that many of these points have already been covered in our responses to your major and minor comments above.

1. are these a sub-type of ischemic heart disease? A non-expert (like me) may be confused;

yes, acute myocardial infarction (AMI) is one of the life threatening manifestations of ischemic heart disease, according to "www.ncbi.nlm.nih.gov/books/NBK209964; Institute of Medicine (US) Committee on Social Security Cardiovascular Disability Criteria. Cardiovascular Disability: Updating the Social Security Listings. Washington (DC): National Academies Press (US); 2010. 7, Ischemic Heart Disease". We have added this information in the revised manuscript.

2. what does this mean? Especially the "view" verb is not clear to me.

To significantly reduce cardiovascular death, patients should do the following:

- 1.) receive thrombolytic therapy, followed by
- 2.) referral for coronary angiography, while seeking for immediate access to
- 3.) percutaneous coronary intervention (PCI).

To clarify this, we have changed the phrase "with a view to" and the sentence now reads "...followed by immediate referral for coronary angiography to a facility where percutaneous coronary intervention is available".

3. here would be a great place to explain better what product would be conducive for treatment, what product for prevention; and what product could be good for both. Maybe to be reprised in discussion where you can explain more.

Please see our response to your major point no. 1 above. We have emphasized that our cells were engineered to simultaneously provide both prevention and treatment when delivered into patients. We have also revised the legend of **revised Fig. 1A**.

4. Do these references publish the sequences of the ScFv? Did you have to manipulate them? Chose the orientation? The linker? As someone who looks into these, this is usually a painful first step that requires lots of guess-work and optimization. If you did not have to do that due to the rigorousness of the published references, would be a good place to acknowledge that. Sometimes you only find sequences buried in a patent supplementary material...

Following your request in major point no. 3, we now provide all sequences used, produced and cited by this study in a separate spreadsheet. Regions for the two scFv-constructs, which were derived from published articles (*Protein Eng Des Sel* (2012) 25(6):295; *Protein Eng Des Sel* (2013) 26(12):773), are annotated accordingly.

5. Is this speculation? Or derived from molecular modeling? Would be good to know as you don't present data to support this

Please refer to our response to minor point no. 4 of reviewer 1. After talking to experts in protein design, we have removed speculative statements on intrachain dimerization and non-functional homodimer formation. The revised paragraph now reads: "Importantly, the linker between both

scFv domains was carefully chosen to be short, so that the distance between the cTnI epitopes remains smaller than the distance between complementarity-determining regions (CDRs). In addition, these homodimeric receptors may show increased sensitivity by potentially binding two cTnI proteins simultaneously, thereby gaining in avidity”.

6. I think it is a good service to the community if we spell out which mammalian cells we are using in the main text.

Not clear at this point from reading this what exactly is the inducer here. Is it from the media? Produced from cells? Extracted? I recommend to include here “Via SEAP reporter assay”

Curious about how this induction strength with your SEAP assay compare to other previously developed receptors: it is remarkable that a new receptor with just swapping the extracellular domain retains high functionality without incurring in reduction of induction fold for example; if that is the case I would highlight it!

We have added all the requested information. We have also followed your suggestion and added remarks on flexible swapping of the extracellular domains: “These results show that functional receptors could be created despite exchanging the extracellular domains, thus highlighting the modular nature of this approach”.

7. This raised the question to me: does it indeed activate endogenous signaling dynamics? Does it interfere/or contribute to the phenotypic outcomes?

Please see our response to your major point no. 2 above.

8. Does the receptor show reversibility or not? If the test is for reversibility, you should have gotten a clear answer to that question. Then maybe is providing tonic signaling at this point [no further increase]? How do you distinguish the two possibilities?

Please see our responses to major point no. 3a of reviewer 1 and your major point no. 5 above. To demonstrate reversibility and as well as “off state”-kinetics of the system, we added a reversibility experiment showing the time course of target gene expression following repeated cTnI stimulation (**new Fig. 3E**). The results show that once the cTnI trigger signal was removed by medium exchange, reporter gene expression rapidly became inactive and there was no significant accumulation to produce “leaky expression” within the next 24 h and until the next cTnI stimulation event (**new Fig. 3E**, red line). From this experiment, we can see that the system is reversible and we would not expect that sustained tonic signaling would even come close to the expression levels of the “real” ON-state, and thus should not cause stimulus-independent therapeutic effects as often reported for other types of cell therapies (*J Immunol* (2019) 202(6):1735).

9. The dataset in Fig. 2C I don’t think it supports a strict stop-and-go activation

Please refer to our response to major point no. 5 above. We have removed the “stop-and-go” term. To better illustrate the dynamics of receptor-mediated gene expression, we have added a reversibility experiment showing strict cTnI-dependent transcriptional activation (**new Fig. 3E**; see our responses to major point no. 3a of reviewer 1 and your previous point above).

10. Why systemic range? Not clear given I am not sure what is the clinical scenario the authors have in mind. The focus on systemic range makes me suspect the authors have in mind a circulating cell therapeutic application. Is that correct? Or is it just to have a ballpark to optimize to?

According to clinical standards (*Circulation* (2011) 124(21):2350), the pathological concentration range of cTnI in human blood is 0.1-100 ng/mL. Thus, CardioProtect must be capable of responding to cTnI within this “systemic range”. This was the goal of the experiment in **Fig. 3C** (old Fig. 3D).

11. My reading is that led to the generation of new receptor constructs. If that is the case the complete sequence of these receptors should be made available as supplementary material.

We have now uploaded all relevant sequences mentioned in our manuscript using a separate spreadsheet (see our responses to major point no. 3 and your in-text comment no. 4 above).

12. Not clear what this means in this context. Is it the shortest that lead to any activation? If so this is not supported, as this is the shorter that is presented (to support minimal claim would need to see shorter exposure times that don't display activation). Or did the authors mean just very short?

This experiment was only designed to establish that a relatively short induction time that can sufficiently activate the system. We have not tested even shorter stimulation pulses (e.g. <5 min) and have accordingly removed the “minimal” claim.

13. Not clear what this means in this context. Is this produced as recombinant protein for medical use? Is it a natural sequence? Or derived from a natural sequence but post-protease processing? Or is it a completely synthetic sequence?

The term “recombinant protein” is defined as any type of therapeutic or “ready-to-use” protein manufactured by recombinant DNA technology (*Cardiovasc Ther* (2012) 30(2):e81). The production can be based on either natural or synthetic sequences introduced into a fermentation-compatible host cell type for large-scale bioprocess engineering. Many thus-defined recombinant thrombolytic proteins (including the tenecteplase (TNK) construct that was chosen for our study) are currently available on the market (*CNS Drugs* (2015) 29(10):811).

14. What is this? A disease? Why not talking a little bit about this in the introduction?

We have added a few remarks on ST-segment elevation myocardial infarction (STEMI) to the

revised introduction.

15. This seems to me a mis-representation of the findings. With these approaches you are just lowering the basal activation. Which yes, obviously, results in increased fold change as you don't affect the induced state. But you don't restore the peak induction capacity. At least you should state this.

Peak induction capacity is still 40 vs 200

If fold changes is what matters clinically, the authors should explain why

To address this question, we have estimated the efficacy window of TNK. First, we added an experimental setup for visualization of clot formation from platelet-rich-plasma (**new Fig. S4A**). Considering the experimental conditions set for fibrinolysis of clots produced from platelet-poor-plasma (**new Fig. S4B**), we reasoned that at least 1-3 ng/mL TNK appeared to be required to observe significant clot lysis throughout all assays (**new Fig. S4**). At the same time, we could observe saturating fibrinolytic effects with 27 ng/mL TNK, indicating that encapsulated CardioProtect can produce sufficient TNK amounts in its fully activated state (40 ng/mL; Fig. 5B), even though non-encapsulated cells had greater production rates (200 ng/mL; Fig. 5B). In other words, whether or not encapsulated, CardioProtect cells can always produce sufficient amounts of TNK in the activated state, while basal or excessive expression can be effectively attenuated if necessary by using the doxycycline-dependent safety switch (**new Fig. S6**), though it's important to note that another experimental dataset from the thrombolysis assay suggest that this baseline expression (both the 0 ng/ml cTnI condition and the dox condition) is in any case incapable of triggering premature clot lysis (**new Fig. S7**). Nevertheless, we agree that the word "restore" may be misleading at this point, and that fold-changes indeed play a minor role in the context of clinical applications. Thus, we have revised the corresponding section by focusing on the efficacy window instead (see also our response to your in-text comment no. 14 above).

16. It would be really helpful if you could explain here a bit more exactly how this system is built, what were your expectations, what were the assumptions of each steps. The description is rather superficial and does not give me a great sense of what exactly is going on

Please refer to our response to your minor point no. 7 above. This experimental setup allows instant visualization of clot lysis triggered by cTnI-mediated TNK release by placing key components, i.e., a) human blood, b) fibrin clots and c) encapsulated "therapeutic" cells in the same reaction system - a scenario that models AMI in humans (**Fig. 5D**). Fig. 4B,4C shows that the fibrin clots generated using our method indeed have similar morphologies, composition and size to those formed after rupture of atherosclerotic plaques in humans, whereas Fig. 4D,4E and **new Fig. S4** confirm that TNK produced by CardioProtect indeed has the capacity to lyse these clots. Furthermore, following your request to show a temporal evolution of the entire thrombolysis process (see our response to your major point no. 7 above), we show that complete clot lysis can be usually observed at 36 h after exposure to TNK (**new Fig. S7**). To improve the data presentation, we have changed the handwritten labels in **Fig. 5E,5F** into electronic fonts and

grouped all replicates for a same experimental condition (see our response to minor point of reviewer 1 above).

17. Very strong paragraph with very sensible explanation of support to your ex-vivo model

Thank you.

18. What would be the clinical manifestation that would trigger the requirement for a re-adjustment of the dose?

In clinical settings, (hyper)activity of implanted cells may be identified in terms of plasma D-dimer levels *in vivo* (**Fig. 5E,5F** and **new Fig. S7**), which would be a critical readout to decide whether or when either re-adjusting the dose or completely shutting down the system with the safety switch (**new Fig. S6**) should be initiated.

19. Figure Legends of Fig. 5E: In a 24-well plate, encapsulated CardioProtect cells were cultivated in human whole blood pre-mixed with an equal volume of RPMI 1640 medium, then a freshly prepared fibrin clot (prepared as described in Fig. 4B) was added together with concentrated inducer solution. Time-dependent clot lysis was monitored over 60 h.

We have revised this sentence by specifying the molecular details of the “concentrated inducer solution” (cTnI and doxycycline doses as indicated in the revised figure labels). Following your advice in major point no. 7, we have recorded the temporal evolution of the corresponding clot lysis process (**new Fig. S7**). Therefore, the 60 h data previously shown in Fig. 5E,5F represent endpoint measurements of this time-dependent clot lysis process, with both datasets showing the same finding that there was no fibrinolytic activity in the basal 0 ng/mL cTnI condition or in the dox condition. Also, to improve the data presentation, we have changed the handwritten labels in Fig. 5E,5F into electronic fonts and grouped all replicates for the same experimental condition (see our response to minor point of reviewer 1 above).

25th Aug 2025

Manuscript Number: MSB-2024-12665R

Title: Engineering mammalian cells for detection and treatment of cardiac injury

Dear Dr. Fussenegger,

Thank you for the submission of your revised manuscript to Molecular Systems Biology. I am pleased to inform you that we will be able to accept your manuscript pending the following final amendments and appropriate response to reviewers:

- 1) There is a name discrepancy between the manuscript and our submission system: in the manuscript there is the author name Bozhidar-Adrian Stefanov vs. in our submission system it is listed as Adrian Stefanov. Please ensure that these match.
- 2) We require an ORCID ID for corresponding authors - a request for the ORCID to be linked in our submission system was sent previously to Dr. Mingqi Xie on July 7th, 2025 with instructions. Please ensure that Dr. Xie has done this prior to resubmission.
- 3) Please provide a functional email address for the following authors, as the given contact email address bounced back messages that were delivered to them: Adrian Stefanov (adrian.stephanov@unibe.dh), Jian Lv (Lvjian@westlake.edu.cn), Zhihua Wang (wangzhihua@westlake.edu.cn)
- 4) In the main manuscript file, please remove the figures and place the Figure Legends below the References. Please also ensure that there are no track changes when resubmitting.
- 5) Please reduce keywords to max. 5.
- 6) As you have no data deposited in external repositories, the Data Availability statement should include the following statement: "This study includes no data deposited in external repositories."
- 7) Please rename "Competing Interests" to "Disclosure and competing interests statement". We updated our journal's competing interests policy in January 2022 and request authors to consider both actual and perceived competing interests. Please review the policy <https://www.embopress.org/competing-interests> and update your competing interests if necessary.
- 8) Author contributions: Please remove it from the manuscript and specify author contributions in our submission system. CRediT has replaced the traditional author contributions section because it offers a systematic machine-readable author contributions format that allows for more effective research assessment. You are encouraged to use the free text boxes beneath each contributing author's name to add specific details on the author's contribution. More information is available in our guide to authors: <https://www.embopress.org/page/journal/17574684/authorguide#authorshipguidelines>
- 9) Please correct the reference citation in the reference list to be alphabetical (not numerical). Where there are more than 10 authors on a paper, only the first 10 should be listed, followed by "et al.". Please check "Author Guidelines" for more information. <https://www.embopress.org/page/journal/17574684/authorguide#referencesformat>
- 10) In the Methods, please take care of the following:
 - The Materials and Methods section should be renamed to "Methods".
 - The Methods should be included in the main manuscript after the Discussion, not the Appendix.
 - Studies with human research participants: Please state that the experiments conformed to the principles set out in the WMA Declaration of Helsinki and the Department of Health and Human Services Belmont Report. Please note that this is a separate statement from the specific ethics committee approval and informed consent.
 - Cell lines: Please be sure to include a sentence in the Methods as to whether or not the cell lines were recently authenticated and tested for mycoplasma contamination. Please also be sure to update the Author Checklist with this information and where it can be found in the manuscript.
 - Please ensure that a statement on whether or not blinding was done is included in the Methods even if no blinding was done. Please also be sure to update the Author Checklist with this information and where it can be found in the manuscript.
- 11) All Materials and Methods need to be described in the main text using our 'Structured Methods' format. According to this format, the Methods section includes a Reagents and Tools Table (listing key reagents, experimental models, software and relevant equipment and including their sources and relevant identifiers) followed by a Methods and Protocols section describing the methods, ideally using a step-by-step protocol format. The aim is to facilitate adoption of the methodologies across labs. Please download and fill our Reagents and Tools Table template (.docx), which you can find in our author guidelines: <https://www.embopress.org/page/journal/14693178/authorguide#structuredmethods>. When submitting your revised manuscript, please do not include the Reagents and Tools Table in the Methods section of the manuscript but upload it as a separate file choosing the file type "Reagent Table". An example of a Method paper with Structured Methods can be found here: <https://www.embopress.org/doi/10.15252/msb.20178071>.
- 12) Please place individual sections of the manuscript in the following order: Title page - Abstract & Keywords - Introduction - Results - Discussion - Methods - Data Availability - Acknowledgements - Disclosure and Competing Interests Statement - References - Figure Legends - Expanded View Figure Legends.
- 13) For the figures and figure legends, please take care of the following:
 - Please remove all figures from main manuscript file and leave only main figure legends placed after the references.
 - Please note that the exact p values are not provided in the legends of figures 3D, 5F

14) Please upload the troponin sequence file as Dataset EV1 and add a legend for the table in the excel file in a separate tab. Please update the callout in the main manuscript text.

15) Appendix file: Please upload the Appendix as a single PDF (no separate image files are needed). The title page should contain "Appendix for + manuscript title" and a Table of Contents with page numbers for the listed items. The nomenclature should be Appendix Figure Sx and Appendix Table Sx throughout manuscript and Appendix PDF.

16) Please ensure that all funding sources are entered into the manuscript submission system. Currently the following are missing in our submission system: Ministry of Science and Technology (MOST Project 2020YFA0909200), the National Natural Science Foundation of China (NSFC Project 32071429), the HRHI program 202209009 of Westlake Laboratory of Life Sciences and Biomedicine, Westlake Education Foundation and Tencent Foundation

17) Synopsis:

- Synopsis image: Please provide a graphic that summarises the main findings of the manuscript on a glance and upload it as a high-resolution jpeg file 550 pixels wide x (300-600) pixels high.

- Synopsis text: Please provide a separate word document including a short standfirst (maximum of 300 characters, including spaces) and up to 5 bullet points to summarise the key NEW findings. They should be designed to be complementary to the abstract - i.e. not repeat the same text. We encourage inclusion of key acronyms and quantitative information (maximum of 30 words / bullet point). Please use the passive voice.

18) Source Data: Source Data should be organized as a single source data file (zipped) per figure for main figures (all EV and/or Appendix figure Source Data can be included in a single folder), with the panels clearly visible in the folder structure instead of a single excel file for all Source Data. e.g. all the Source data files for figure 1 need to be saved in a single folder and this needs to be zipped and then uploaded as "SD figure 1.zip" file.

19) As part of the EMBO Publications transparent editorial process initiative (see our policy here:

https://www.embopress.org/transparent-process#Review_Process), Molecular Systems Biology will publish online a Peer Review File (PRF) to accompany accepted manuscripts. This file will be published in conjunction with your paper and will include the anonymous referee reports, your point-by-point response and all pertinent correspondence relating to the manuscript. Let us know whether you agree with the publication of the PRF and as here, if you want to remove or not any figures from it prior to publication. Please note that the Authors checklist will be published at the end of the PRF.

20) After your paper is published, we may promote it on social media. If you have any handles or hashtags for Bluesky you would like included, please let us know.

21) Please provide a point-by-point letter INCLUDING my comments as well as the reviewer's reports and your detailed responses (as Word file).

I look forward to reading a new revised version of your manuscript as soon as possible.

Yours sincerely,

Poonam Bheda, PhD
Scientific Editor
Molecular Systems Biology

Reviewer #1:

Dear Editor,

I am pleased to confirm that the authors have adequately addressed my previous concerns through additional experiments and clarifications provided throughout the manuscript. I am happy to recommend the manuscript for publication.

Reviewer #3:

The revision of the manuscript by Yaqing Si and colleagues has substantially improved overall an already impactful manuscript. The authors improved significantly the clarity of the presentation and addressed most of the reviewers comment by adding new experimental data.

Only a (minor) outstanding comment remains among my comments, regarding the main comment #2, where I was asking: "Some of the receptor constructs activate, at least in terms of design, endogenous cell pathways alongside the synthetic target gene. Is that true? I don't see experiments to test that in the current manuscript. Is that not relevant? Why not? Could some of the phenotypic outcomes attributed to synthetic gene activation be attributable to change in cell physiology downstream of activation of endogenous pathways? Either way is fine, would be good for the authors to guide the reader to interpret these nuances of the hybrid synthetic receptor design and applications."

The authors responded with new experiments that convincingly show that other endogenous pathways that go through Stat or ERK activation are not affected by the expression of the synthetic hybrid receptors. This addresses one of my concerns. There is another side to this though that I did not see addressed (maybe it was not clear from my reviewer comment): when you activate your synthetic hybrid receptors to induce TNK production, you also activate other STAT and ERK endogenous target genes, alongside your genetically engineered one: does that by itself do something to the clots? This question would be addressed by an experiment with the synthetic receptor cells that activate SEAP instead of the therapeutic target genes. Have you ever tried to see if the induction of a synthetic hybrid receptor/reporter cassette has an effect in clotting assays? If you have done such experiments (and they are not in the current version of the manuscript), would be good to share them with the community to strengthen the point that the effects you see with this hybrid synthetic receptors is purely due to the activation of an exogenous target gene, and not to the activation of endogenous responses.

I do want to say, it is more the type of "would be good to know" information, I don't think it is crucial for the paper to stand as it is by itself. I am just sharing as I liked this work, and I think this would provide a completion to its scope, in my opinion.

MSB-2024-12665R: point-by-point letter

- 1. There is a name discrepancy between the manuscript and our submission system: in the manuscript there is the author name Bozhidar-Adrian Stefanov vs. in our submission system it is listed as Adrian Stefanov. Please ensure that these match.**

We've corrected the name to Bozhidar-Adrian Stefanov in submission system.

- 2. We require an ORCID ID for corresponding authors - a request for the ORCID to be linked in our submission system was sent previously to Dr. Mingqi Xie on July 7th, 2025 with instructions. Please ensure that Dr. Xie has done this prior to resubmission.**

Mingqi Xie's ORCID ID has been uploaded in the submission system.

- 3. Please provide a functional email address for the following authors, as the given contact email address bounced back messages that were delivered to them: Adrian Stefanov (adrian.stephanov@unibe.dh), Jian Lv (Lvjian@westlake.edu.cn), Zhihua Wang (wangzhihua@westlake.edu.cn)**

Sorry for that, their functional emails are provided below:

- Bozhidar-Adrian Stefanov: adrian.stephanov@unibe.ch
- Jian Lv: lvjian@fuwaisz.cn
- Zhihua Wang: wangzhihua@fuwaihospital.org

- 4. In the main manuscript file, please remove the figures and place the Figure Legends below the References. Please also ensure that there are no track changes when resubmitting.**

Figures have been removed, and the legends were placed below the Reference. Though, we display specific changes in the main text document in red fonts.

- 5. Please reduce keywords to max. 5.**

Done as requested. Heart disease and synthetic biology were removed.

6. As you have no data deposited in external repositories, the Data Availability statement should include the following statement: "This study includes no data deposited in external repositories."

This statement has been added into Data Availability Statement.

7. Please rename "Competing Interests" to "Disclosure and competing interests statement".

Corrected.

8. Author contributions: Please remove it from the manuscript and specify author contributions in our submission system.

Author contributions have been removed and re-filled in submission system.

9. Please correct the reference citation in the reference list to be alphabetical (not numerical). Where there are more than 10 authors on a paper, only the first 10 should be listed, followed by "et al.". Please check "Author Guidelines" for more information.

Reference citation has been corrected as alphabetical format as requested.

10. In the Methods, please take care of the following:

- The Materials and Methods section should be renamed to "Methods".
- The Methods should be included in the main manuscript after the Discussion, not the Appendix.
- Studies with human research participants: Please state that the experiments conformed to the principles set out in the WMA Declaration of Helsinki and the Department of Health and Human Services Belmont Report. Please note that this is a separate statement from the specific ethics committee approval and informed consent.

- **Cell lines: Please be sure to include a sentence in the Methods as to whether or not the cell lines were recently authenticated and tested for mycoplasma contamination. Please also be sure to update the Author Checklist with this information and where it can be found in the manuscript.**
- **Please ensure that a statement on whether or not blinding was done is included in the Methods even if no blinding was done. Please also be sure to update the Author Checklist with this information and where it can be found in the manuscript.**

Materials and Methods section has been renamed as 'Methods' and remains in the main text. The statement of adhering to the principles set out in the WMA Declaration of Helsinki and the Department of Health and Human Services Belmont Report has been added in 'Thrombolysis assay' part; Cell lines authentication and clear of mycoplasma clarification have been added in 'Cell Culture and Transfection' part as well as in the Author Checklist. A statement "all investigators were blinded to the treatment conditions" was added to the Methods section (Animal experiments) as well as in the Author Checklist.

- 11. All Materials and Methods need to be described in the main text using our 'Structured Methods' format. According to this format, the Methods section includes a Reagents and Tools Table (listing key reagents, experimental models, software and relevant equipment and including their sources and relevant identifiers) followed by a Methods and Protocols section describing the methods, ideally using a step-by-step protocol format. The aim is to facilitate adoption of the methodologies across labs.**

Please download and fill our Reagents and Tools Table template (.docx), which you can find in our author guidelines:

All materials and methods have been described in the main text using 'Structured Methods' format according to the guidelines. Reagents and Tools are now provided in a separate file as requested (MSB-2024-12665-Reagents_Tools_Table.docx).

- 12. Please place individual sections of the manuscript in the following order:**
Title page - Abstract & Keywords - Introduction - Results - Discussion - Methods - Data Availability - Acknowledgements - Disclosure and Competing Interests Statement - References - Figure Legends - Expanded View Figure Legends.

The order of the whole paper has been adjusted as requested.

- 13. For the figures and figure legends, please take care of the following:**
- Please remove all figures from main manuscript file and leave only main figure legends placed after the references.
 - Please note that the exact p values are not provided in the legends of figures 3D, 5F

Figures have been removed from main manuscript. The exact p values have been provided in the legends of figures 3D, 5F.

- 14. Please upload the troponin sequence file as Dataset EV1 and add a legend for the table in the excel file in a separate tab. Please update the callout in the main manuscript text.**

Troponin sequence file will be uploaded as **Dataset EV1.xlsx**, which is now cited by the Methods section of the main text (Vector Design part).

- 15. Appendix file: Please upload the Appendix as a single PDF (no separate image files are needed). The title page should contain "Appendix for + manuscript title" and a Table of Contents with page numbers for the listed items. The nomenclature should be Appendix Figure Sx and Appendix Table Sx throughout manuscript and Appendix PDF.**

Appendix has been uploaded as a single PDF, title page contains "Appendix for + manuscript title" and a Table of Contents with page numbers for the listed

items. Appendix Figure Sx and Appendix Table Sx were cited throughout manuscript and Appendix PDF.

- 16. Please ensure that all funding sources are entered into the manuscript submission system. Currently the following are missing in our submission system: Ministry of Science and Technology (MOST Project 2020YFA0909200), the National Natural Science Foundation of China (NSFC Project 32071429), the HRHI program 202209009 of Westlake Laboratory of Life Sciences and Biomedicine, Westlake Education Foundation and Tencent Foundation**

These fundings have been entered into the manuscript submission system.

17. Synopsis:

- **Synopsis image: Please provide a graphic that summarises the main findings of the manuscript on a glance and upload it as a high-resolution jpeg file 550 pixels wide x (300-600) pixels high.**
- **Synopsis text: Please provide a separate word document including a short standfirst (maximum of 300 characters, including spaces) and up to 5 bullet points to summarise the key NEW findings. They should be designed to be complementary to the abstract - i.e. not repeat the same text. We encourage inclusion of key acronyms and quantitative information (maximum of 30 words / bullet point). Please use the passive voice.**
- **Please check your synopsis text and image before submission with your revised manuscript. Please be aware that in the proof stage minor corrections only are allowed (e.g., typos).**

An individual synopsis file (.docx) containing a high-resolution JPEG will be uploaded.

- 18. Source Data: Source Data should be organized as a single source data file (zipped) per figure for main figures (all EV and/or Appendix figure Source Data can be included in a single folder), with the panels clearly visible in the folder structure instead of a single excel file for all Source Data. e.g. all the**

Source data files for figure 1 need to be saved in a single folder and this needs to be zipped and then uploaded as "SD figure 1.zip" file.

Source Data has been organized as requested.

19. As part of the EMBO Publications transparent editorial process initiative (see [our policy here: https://www.embopress.org/transparent-process#Review Process](https://www.embopress.org/transparent-process#Review_Process)), Molecular Systems Biology will publish online a Peer Review File (PRF) to accompany accepted manuscripts. This file will be published in conjunction with your paper and will include the anonymous referee reports, your point-by-point response and all pertinent correspondence relating to the manuscript. Let us know whether you agree with the publication of the PRF and as here, if you want to remove or not any figures from it prior to publication. Please note that the Authors checklist will be published at the end of the PRF.

We agree with the publication of RPF.

20. After your paper is published, we may promote it on social media. If you have any handles or hashtags for Bluesky you would like included, please let us know.

Thank you for your consideration, we don't have such need so far.

Reviewer's comment:

The authors responded with new experiments that convincingly show that other endogenous pathways that go through Stat or ERK activation are not affected by the expression of the synthetic hybrid receptors. This addresses one of my concerns. There is another side to this though that I did not see addressed (maybe it was not clear from my reviewer comment): when you activate your synthetic hybrid receptors to induce TNK production, you also activate other STAT and ERK endogenous target genes, alongside your genetically engineered

one: does that by itself do something to the clots? This question would be addressed by an experiment with the synthetic receptor cells that activate SEAP instead of the therapeutic target genes. Have you ever tried to see if the induction of a synthetic hybrid receptor/reporter cassette has an effect in clotting assays? If you have done such experiments (and they are not in the current version of the manuscript), would be good to share them with the community to strengthen the point that the effects you see with this hybrid synthetic receptors is purely due to the activation of an exogenous target gene, and not to the activation of endogenous responses.

I do want to say, it is more the type of "would be good to know" information, I don't think it is crucial for the paper to stand as it is by itself. I am just sharing as I liked this work, and I think this would provide a completion to its scope, in my opinion.

First of all, we thank this reviewer for his/her supportive and constructive comments through the entire revision process. Indeed, as correctly pointed out by this reviewer, our synthetic receptors were designed to simultaneously activate endogenous signaling pathways, such as MAPK/ERK or JAK/STAT3 (**Figs. S2,S3**). To discuss whether activation of endogenous target genes could contribute to therapeutic effects, we believe that the data presented in **Fig. 5E** are most representative. Here, the experimental group of simultaneously adding doxycycline and cTnI would initiate endogenous MAPK signaling, while shutting off any synthetic MAPK-dependent promoters. In this state, endogenous MAPK signaling in CardioProtect would remain fully activated due to the constant presence of cTnI. Though, no fibrinolysis effects were observed under this condition, suggesting that activation of only endogenous MAPK target genes would be insufficient to induce therapeutic (side)-effects. We have added this point to the revised manuscript (Results section).

26th Sep 2025

Manuscript number: MSB-2024-12665RR

Title: Engineering mammalian cells for detection and treatment of cardiac injury

Dear Dr. Fussenegger,

Thank you again for sending us your revised manuscript. We are now satisfied with the modifications made and I am pleased to inform you that your paper has been accepted for publication.

Yours sincerely,

Sincerely,

Poonam Bheda, PhD
Scientific Editor
Molecular Systems Biology
